# THE INTRICATE DANCE OF PROMPT COMPLEXITY, QUALITY, DIVERSITY, AND CONSISTENCY IN T2I MODELS

**Zhang Xiaofeng**[1,2,3]**, Aaron Courville**[1,2,5]**, Michal Drozdzal**[3]**, Adriana Romero-Soriano**[1,3,4,5]
[1] Mila - Québec AI Institute [2] Université de Montréal [3] FAIR at Meta - Montréal
[4] McGill University [5] Canada CIFAR AI chair
xiaofeng.zhang@mila.quebec

## ABSTRACT

Text-to-image (T2I) models offer great potential for creating virtually limitless synthetic data, a valuable resource compared to fixed and finite real datasets. Previous works evaluate the utility of synthetic data from T2I models on three key desiderata: quality, diversity, and consistency. While prompt engineering is the primary means of interacting with T2I models, the systematic impact of prompt complexity on these critical utility axes remains underexplored. In this paper, we first conduct synthetic experiments to motivate the difficulty of generalization *w.r.t.* prompt complexity and explain the observed difficulty with theoretical derivations. Then, we introduce a new evaluation framework that can compare the utility of real data and synthetic data, and present a comprehensive analysis of how prompt complexity influences the utility of synthetic data generated by commonly used T2I models. We conduct our study across diverse datasets, including CC12M, ImageNet-1k, and DCI, and evaluate different inference-time intervention methods. Our synthetic experiments show that generalizing to more general conditions is harder than the other way round, since the former needs an estimated likelihood that is not learned by diffusion models. Our large-scale empirical experiments reveal that increasing prompt complexity results in lower conditional diversity and prompt consistency, while reducing the synthetic-to-real distribution shift, which aligns with the synthetic experiments. Moreover, current inference-time interventions can augment the diversity of the generations at the expense of moving outside the support of real data. Among those interventions, prompt expansion, by deliberately using a pre-trained language model as a likelihood estimator, consistently achieves the highest performance in both image diversity and aesthetics, even higher than that of real data. Combining advanced guidance interventions with prompt expansion results in the most appealing utility trade-offs of synthetic data.[1]

## 1 INTRODUCTION

Text-to-image (T2I) models have made significant progress in recent years, enabling the generation of high-quality images from textual descriptions (Esser et al., 2024; Labs, 2024). These advancements have opened up new opportunities in several application domains, making it possible for the public to create images easily with their own words. For the research community, the success of T2I models has also unlocked the exploration of synthetic data generated by these models for downstream model training (Hemmat et al., 2024; Askari-Hemmat et al., 2025; Tian et al., 2023; Fan et al., 2024; Dall'Asen et al., 2025) and model self-improvement (Yoon et al., 2024).

Recent studies have yielded in-depth analyses on the utility of synthetic data from off-the-shelf T2I models (Astolfi et al., 2024; Hall et al., 2024c; Lee et al., 2023). The utility of synthetic data from conditional image generative models, such as T2I, has been defined as a set of desiderata (Astolfi

---

[1]Code available at https://github.com/facebookresearch/synthetic_data_utility_prompt_complexity

et al., 2024; DeVries et al., 2019), including *synthetic image quality –i.e.*, aesthetics and realism–, *diversity*, and *conditional consistency –i.e.*, alignment with the conditioning prompt. Analyses of these desiderata have revealed that the impressive progress in image quality has come at the expense of generation diversity. As a result, the community has devoted significant work to devise inference-time interventions –*e.g.*, prompt rewriting (Datta et al., 2024) and advanced guidance approaches modifying standard classifier-free guidance (CFG) (Ho & Salimans, 2022) – that improve the diversity of synthetic data. Although these interventions focus on either changing the prompt conditioning or the flow of signals from the conditioning, *the effect of the prompt content on the utility of synthetic data remains an open question*. Prompts may be characterized by their complexity, defined as the amount of details contained in the prompt or the specificity of its concepts, and state-of-the-art T2I models have been shown to benefit from synthetic and descriptive captions, generated by vision language models (VLMs), during training to improve their generation performance (Betker et al., 2023). It is now common practice to train high performance T2I models on a combination of real and synthetic captions (Esser et al., 2024; Chen et al., 2023; Wu et al., 2025). Yet, these models are known to produce poor results when sampled out of their training distribution (Betker et al., 2023). Considering the very descriptive and synthetic captions used to train T2I models, evaluating these models from the perspective of prompt complexity is crucial to better understand the models' performance.

Therefore, in this paper, we study the effect of prompt complexity on the different utility axes of synthetic image data. We start by conducting synthetic experiments over mixtures of Gaussians, conditioning the generation process on prompts of varying complexities.We show that generalization across prompt complexities is already hard in this toy synthetic setting, especially when attempting to generalize to more general prompts –*i.e.*, prompts that are shorter or less specific than the ones used for training. The results from the synthetic setting motivate an in-depth evaluation of how prompt complexity influences different utility axes of synthetic data from T2I models. To do so, we propose a novel evaluation framework that constructs prompts with different complexities, enabling the evaluation from a prompt complexity perspecitve over a wide range of commonly used vision and vision-language datasets –*i.e.*, CC12M (Changpinyo et al., 2021), ImageNet-1k (Deng et al., 2009), and Densely Captioned Images (DCI) (Urbanek et al., 2024). We use the resulting prompts to condition the generation process of state-of-the-art T2I models –*i.e.*, LDMv1.5 (Rombach et al., 2022), LDM-XL (Podell et al., 2024), LDMv3.5M (Esser et al., 2024), LDMv3.5L (Esser et al., 2024), Flux-schnell (Labs, 2024), and Infinity (Han et al., 2025)–, and collect synthetic images with several inference-time intervention methods including CFG (Ho & Salimans, 2022), condition-annealing diffusion sampling (CADS) (Sadat et al., 2024), interval guidance (Kynkäänniemi et al., 2024), adapted projected guidance (APG) (Sadat et al., 2025), and prompt expansion (Datta et al., 2024). Our analysis unlocks comparisons between the utility of real and synthetic data, which are beneficial to identify potential gaps between real and synthetic image distributions. To the best of our knowledge, *we are the first to systematically evaluate the effect of prompt complexity on T2I generations*.

Our synthetic experiments show that generalizing to more general conditions is harder than generalizing to more fine-grained conditionings, since the former needs an estimated likelihood that is not learned by diffusion models. Our large-scale empirical evaluation enabled by the contributed framework further reveals interesting findings as follows: 1) The trend of utilities are non-linear, and especially the aesthetic score exhibits a slope steeper towards shorter prompt lengths and more gradual for longer ones (Fig. 3c), showing an asymmetry of prompt length generalization. 2) Diversity does not collapse but plateaus as prompt length increases, as shown in Fig. 2c. This suggests an inherent "lower bound of diversity" in T2I models. 3) Optimizing for reference-free metrics harms distributional fidelity. As shown in Figs. 5 and 13, prompt expansion degrades precision and density while newer models (e.g., LDMv3.5L) degrade in frechet distance with real data. This suggests a cautious usage of synthetic data generated from T2I models for downstream applications. 4) By combining advanced guidance methods (in particular APG) with prompt expansion, we can benefit from the advantages of both approaches and achieve the most interesting trade-offs. Overall, our study suggests that prompt complexity is a crucial axis to consider when prompting T2I models, and requires more investigation especially when generating from very general prompts. In addition, diversity is a key feature of real-world image distributions, which is still not properly captured by synthetic images from state-of-the-art T2I models when no explicit prompt expansion is conducted during inference.

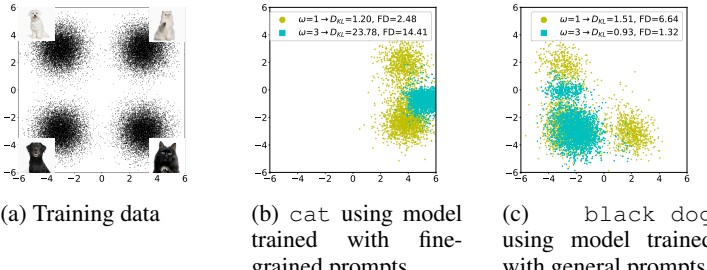

(a) Training data

(b) `cat` using model trained with fine-grained prompts

(c) `black dog` using model trained with general prompts

Figure 1: **Generalization to prompts of different complexities during inference.** 1a shows the training data distribution. 1b presents the generated samples using the general prompt `cat` with the model trained with fine-grained prompts. 1c shows the generated samples using the fine-grained prompt `black dog` with the model trained with general prompts. $\omega$ is the classifier-free guidance scale. With $\omega > 1$, generalization towards more general prompts is harder in this synthetic setting.

## 2  GENERALIZING TO GENERAL CONDITIONS IS HARD

In this section, we use a synthetic setting to build intuition of how T2I models may perform when evaluated on prompts of different complexities, and notably when evaluated on prompts that are outside of their training distribution. motivating the necessity of conducting systematic evaluations of T2I models from a prompt complexity perspective in the following sections.

**Dataset and model.**     Let our data be a mixture of four Gaussians (Figure 1a), where each Gaussian represents a different category –*e.g.*, white dog, white cat, black dog, and black cat. We train two conditional U-Net models (Ronneberger et al., 2015; Rombach et al., 2022) using two sets of conditional prompts: one model is trained with fine-grained prompts (`white dog`, `white cat`, `black dog`, and `black cat`); the other model is trained with general prompts (`dog`, `cat`, `white` and `black`). We train both models, including their vocabulary dictionary, from scratch using DDPM schedule (Ho et al., 2020). We perform inference in a generalization setting, –*i.e.*, if the model is trained with general prompts, inference is done with fine-grained prompts, and vice versa. We also employ classifier-free guidance (CFG) (Ho & Salimans, 2022) with a guidance scale $\omega$; $\omega = 1$ refers to the conditional model, –*i.e.*, not using classifier-free guidance–, and $\omega > 1$ refers to higher CFG guidance strengths. More details can be found in the Appendix A.1.

**Mathematical derivations.**     Conditioning on general prompts is analogous to applying an `OR` operator on fine-grained prompts –*e.g.*, `dog` would be `white dog OR black dog`–, while conditioning on fine-grained prompts may be seen as performing an `AND` operator among general prompts. Following Liu et al. (2022); Du et al. (2023), we have the following derivations. Given a general prompt $c_g$ that can be decomposed into various independent fine-grained prompts $c_f^i, i \in \{1, 2, ..., K\}$, computing the score function $s_\theta(x_t|c_g)$ at timestep $t$ requires both the score functions and the conditional likelihood $p_\theta(x_t|c_f^i)$ of fine-grained ones. The latter is not available from a diffusion model, as shown in Equation 1. However, given a fine-grained prompt $c_f$ that is composed of several general concepts $c_g^i, i \in \{1, 2, ..., M\}$, the score function $s_\theta(x_t|c_f)$ at timestep $t$ can be approximately estimated by the score functions of general prompts solely (*e.g.*, `white` and `dog`), as shown in Equation 2. Derivations are in Appendix A.2.

$$\text{OR operator: } s_\theta(x_t|c_g) = \sum_{i \in \{1,2,...,K\}} \left( \underbrace{\frac{p_\theta(x_t|c_f^i)}{\sum_{j \in \{1,2,...,K\}} p(x_t|c_f^j)}}_{\text{not learned by the diffusion model}} s_\theta(x_t|c_f^i) \right) \tag{1}$$

$$\text{AND operator: } s_\theta(x_t|c_f) = s_\theta(x_t) + \sum_{i \in \{1,2,...,M\}} (s_\theta(x_t|c_g^i) - s_\theta(x_t)) \tag{2}$$

**Generalizing to general prompts is hard.** We use forward KL-divergence ($D_{\text{KL}}$) and Fréchet Distance (FD) to evaluate the generated samples. We also measure reference-free sample diversity using Vendi score (Friedman & Dieng, 2023) (VS)[2], following the empirical evaluation in section 4. On the one hand, Figure 1b shows the generated samples using the general prompt `cat` at inference time with the model trained with fine-grained prompts. When generalizing to general prompts (`OR` operator), the model tends to generate samples towards the mean of the referenced distributions. In particular, when $\omega > 1$ (a common practice in T2I models), the generated samples cover a region where the training data density is very small, resulting in $D_{\text{KL}} = 23.78$, FD $= 14.41$, and $\text{VS}_{\text{gen}} = 1.03$ compared to $\text{VS}_{\text{ref}} = 1.82$. Yet, when $\omega = 1$, this phenomenon is reduced, achieving $D_{\text{KL}} = 1.20$, FD $= 2.48$, and $\text{VS}_{\text{gen}} = 1.43$ compared to $\text{VS}_{\text{ref}} = 1.82$. Since diffusion models only learn the score function (and not the likelihood weighting in Equation 1, they may naively add up the score function values of fine-grained conditions together when generalizing to a general condition, hence disregarding the likelihood-based weighting, and leading to generated samples that correspond to the average of the fine-grained conditional training samples. On the other hand, Figure 1c shows the generated samples using fine-grained prompts at inference time with the model trained with general prompts only. In this case, we observe that the model can leverage both general prompts and generate compositional outcomes in a zero-shot manner. The generated samples achieve $D_{\text{KL}} = 1.51$, FD $= 6.64$, and $\text{VS}_{\text{gen}} = 2.04$ compared to $\text{VS}_{\text{ref}} = 1.10$, when using $\omega = 1$. Note that Equation 2 is similar to CFG, $s_\theta(x_t) + \omega(s_\theta(x_t|c) - s_\theta(x_t))$, when $\omega \approx M$. Empirically, using a larger $\omega$ pushes the distributions further towards the fine-grained conditional direction, reaching $D_{\text{KL}} = 0.93$, FD $= 1.32$, and $\text{VS}_{\text{gen}} = 1.33$ compared to $\text{VS}_{\text{ref}} = 1.10$ when $\omega = 3$. We do not observe severe distributional shift and diversity reduction in this simple synthetic setting. We further discuss the relationship between synthetic settings and large-scale evaluations in Appendix A.3.

## 3 BENCHMARKING FRAMEWORK

We propose a new evaluation framework designed to evaluate the utility (focusing on quality, diversity, and consistency) of synthetic data generated by T2I models as a function of prompt complexity, comparing with that of real data. Comparing the diversity of synthetic images conditioned on a prompt to that of real images in a dataset is challenging, as image-caption pairs in existing datasets are fixed, making it non-trivial to assemble a set of real images that matches a given prompt.

Given an image dataset $\mathcal{X} \subseteq \mathcal{I} \times \mathcal{Y}$, where $\mathcal{I}$ is the image set and $\mathcal{Y}$ is the label set. Each datapoint $X_i \in \mathcal{X}, i \in \{1, 2, \ldots, |\mathcal{X}|\}$ is $(I_i, y_i)$ where $I_i$ represents the image sample, and $y_i$ represents the associated label (either class or captions in our experiments). We first synthesize the images using captions with $K$ different levels of complexities. The caption set associated with each complexity is noted as $\mathcal{C}^k, k \in \{1, 2, \ldots, K\}$. We then use T2I models to generate synthetic images $\bar{\mathcal{I}}^k$ from the caption sets $\mathcal{C}^k$. Finally, we employ different evaluation functions $f \in \mathcal{F} : \{\mathcal{I}, \bar{\mathcal{I}}^k\} \times \mathcal{C}^k \to \mathbb{R}$ to evaluate the utilities of both synthetic and real data, where $\mathcal{F}$ is the set of evaluation functions considered in our analysis and $f$ is an evaluation function. In the following, we describe our framework, which consists of captioning, pairing, alignment, sampling, and generation steps, in detail.

**Captioning.** This step creates captions of different complexities for each datapoint $X_i$. Given the complexity level $K$ that we want to consider in our study, we create captions $c_i^k, i \in \{1, 2, \ldots, |\mathcal{X}|\}, k \in \{1, 2, \ldots, K\}$ of increasing complexities from each datapoint $(X_i, y_i)$. Thus, we transform the original dataset $\mathcal{X}$ into $K$ different datasets $\tilde{\mathcal{X}}^k, k \in \{1, 2, \ldots, K\}$. The dataset $\tilde{\mathcal{X}}^k$ contains all the images in $\mathcal{I}$ and each image $I_i$ is paired with a caption of complexity level-$k$, $c_i^k$.

**Pairing.** Given a certain complexity level $k \in \{1, 2, \ldots, K\}$ and a caption $c_i^k, i \in \{1, 2, \ldots, |\mathcal{X}|\}$, we search for images that are semantically similar to the caption $c_i^k$. We note these images as a set $\bar{\mathcal{I}}^{ik}$ with elements $I_j^{ik}, j \in \{1, 2, \ldots, |\mathcal{X}|\}$. We only keep the image sets $\bar{\mathcal{I}}^{ik}$ with cardinality $\geq 20$. Thus, across complexity levels, the captions left after filtering are different. We denote the indices of the captions left in each complexity as sets $\mathcal{N}_1^{\text{P}}, \mathcal{N}_2^{\text{P}}, \ldots, \mathcal{N}_k^{\text{P}}$.

**Alignment.** Across complexities, the images left in each complexity $\bigcup_{i \in \mathcal{N}_k^{\text{P}}} \bar{\mathcal{I}}^{ik}, k \in \{1, 2, \ldots, K\}$ could be very different due to the filtering performed in the pairing step. This step

---

[2]For the synthetic setting, there is no pre-trained models to evaluate the reference-free quality and consistency. KL-divergence and FD cover both the quality and consistency.

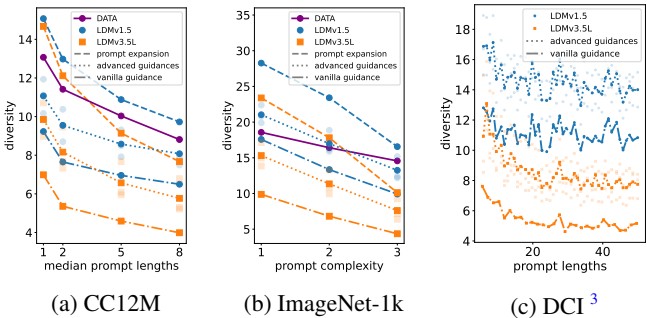

(a) CC12M     (b) ImageNet-1k     (c) DCI [3]

Figure 2: **Reference-free diversity metric.** Diversity (Vendi) of LDMv1.5 and LDMv3.5L generations with CC12M and ImageNet-1k prompts when using: 1) vanilla guidance (CFG), 2) prompt expansion, and 3) advanced guidance methods, for which transparent markers correspond to different methods and the solid marker is the average over methods. Both advanced guidance methods and prompt expansion lead to improved diversity over the vanilla guidance. Prompt expansion from shorter captions can surpass the real data diversity. We further extend to much longer DCI prompts. Diversity of all models first decreases then plateaus which is not observed within shorter prompt length ranges.

aligns the image modality across complexities to ensure comparability. Specifically, we iteratively remove images that are not shared across complexities, *i.e.*, not in the set $\bigcap_{k\in\{1,2,...,K\}} \bigcup_{i\in\mathcal{N}_k^p} \bar{\mathcal{I}}^{ik}$. This process is repeated until the following criteria is satisfied, $\bigcap_{k\in\{1,2,...,K\}} \bigcup_{i\in\mathcal{N}_k^p} \bar{\mathcal{I}}^{ik} = \bigcup_{k\in\{1,2,...,K\}} \bigcup_{i\in\mathcal{N}_k^p} \bar{\mathcal{I}}^{ik}$. We denote the indices of the captions left in each complexity as sets $\mathcal{N}_1^a, \mathcal{N}_2^a, ..., \mathcal{N}_k^a$.

**Sampling.** Following the alignment step, the number of remaining captions may still be too large for practical evaluation of T2I models. We randomly sample the same number captions per complexity in order to maximize semantic coverage in a consistent manner. This results in sets of indices of captions $\mathcal{N}_1^s, \mathcal{N}_2^s, ..., \mathcal{N}_k^s$ where $|\mathcal{N}_k^s|$ is the same $\forall k \in \{1, 2, \ldots, K\}$. The sampled captions are then used as prompts to collect synthetic data from T2I models.

**Generation.** For each prompt (caption) $c_i^k, i \in \mathcal{N}_k^s, k \in \{1, 2, \ldots, K\}$, we generate as many images as the cardinality of the smallest similar image set across different caption complexity $N_{\text{gen}} = \min_{i\in\mathcal{N}_k^s, k\in\{1,2,...,K\}} |\bar{\mathcal{I}}^{ik}|$, ensuring representativeness. The generated image set for prompt $c_i^k$ is noted as $\hat{\mathcal{I}}^{ik}$. With the prompt conditional image sets for both real images and synthetic images, we comprehensively assess each axis of data utility using a suite of representative reference-free metrics, which do not constrain the analysis to any predefined target data distribution. Our framework can also be readily extended to incorporate reference-based evaluation metrics as needed, as shown in section 4.

# 4 EVALUATION OF SYNTHETIC DATA GENERATED FROM T2I MODELS

## 4.1 EXPERIMENTAL SETUP

**Datasets.** We employ three datasets –CC12M, ImageNet-1k, and DCI– to evaluate the utility of synthetic data generated from T2I models. CC12M (Changpinyo et al., 2021) is a large-scale, internet-crawled image-text dataset containing 12 million image-caption pairs, where we investigate the effect of increasing prompt detail and length on the utility axes of synthetic images. ImageNet-1k (Deng et al., 2009) is a widely used, object-centric, curated dataset with 1,000 classes, where we examine the effect of concept specificity on the utility axes. DCI (Urbanek et al., 2024) is a large-scale, human-curated image-caption dataset featuring detailed, long captions for each image, where we further explore prompt complexity using *very long* prompts. More details are presented in Appendix C.1.

---

[3] We stop at prompt length of 50, given the 77 token limit of text encoders of LDMv1.5. For LDMv3.5 models, we push the prompt length to 100, as shown in Figure 11, Appendix D.3.

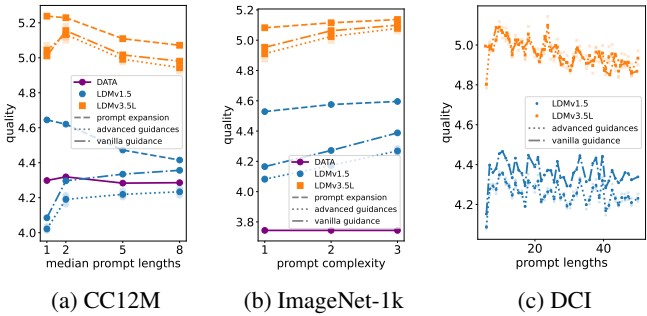

(a) CC12M  (b) ImageNet-1k  (c) DCI

Figure 3: **Reference-free quality metric.** Quality (aesthetic) of LDMv1.5 and LDMv3.5L generations with CC12M and ImageNet-1k prompts when using: 1) vanilla guidance (CFG), 2) prompt expansion, and 3) advanced guidance methods, for which transparent markers correspond to different methods and the solid marker is the average over methods. Advanced guidance methods slightly impair the quality while prompt expansion increases it. We further extend to much longer DCI prompts. Quality of all models first increases then decreases which is not observed within shorter prompt ranges.

**Models.** We select LDMv1.5 (Rombach et al., 2022) and LDMv3.5L (Esser et al., 2024) models for the presentation of results. We also include the evaluation results from LDMv3.5M (Esser et al., 2024) and LDM-XL (Podell et al., 2024) in Appendix D. LDMv1.5 is the early high-performing T2I model using the latent diffusion model (LDM) architecture (Rombach et al., 2022), while LDMv3.5L is its latest successor, employing a rectified flow model (Liu et al., 2023). These models are representative of current open-source T2I models and illustrate the trend of model improvement over time. We consider the following guidance techniques: CFG (Ho & Salimans, 2022) (referred to as vanilla guidance), and advanced guidance approaches including CADS (Sadat et al., 2024), interval guidance (Kynkäänniemi et al., 2024), and APG (Sadat et al., 2025). We also explore the effect of explicitly expanding prompts by adding information, using large language models to expand each caption to $N_{gen}$ different captions with maximum thirty words. Additionally, we report evaluation results of Flux-schnell (Labs, 2024) and Infinity (Han et al., 2025) to cover more model types, to which advanced guidance methods are not applicable. Considering the limited inference-time interventions, we present the results for these two models in Appendix F. More implementation details are presented in Appendix C.

**Metrics.** We use the aesthetic score (discus0434 & Goswami, 2023) to assess image quality, the Vendi score (Friedman & Dieng, 2023) to measure diversity, and the Davidsonian Scene Graph (DSG) score (Cho et al., 2024) to evaluate prompt consistency. Our human evaluations in Appendix E confirm the validity of the automatic metrics. We also report widely used reference-based metrics, including FDD (Fréchet distance using DINOv2 (Oquab et al., 2023)) (Stein et al., 2023), precision (Kynkäänniemi et al., 2019), density and coverage (Naeem et al., 2020). [4]

## 4.2 UTILITY AXES MEASURED BY REFERENCE-FREE METRICS

**Diversity.** We present results in Figure 2. Our evaluation findings empirically verify that the diversity of synthetic data decreases as we increase prompt complexity for all models and interventions considered (Figures 3a and 3b). As increasing the complexity (either by adding details in CC12M or increasing the class specificity in ImageNet-1k) constrains the generation process, the degree of freedom of the model is reduced. In terms of inference-time interventions, both advanced guidance methods and prompt expansion lead to higher diversity than vanilla guidance. Note that prompt expansion can be considered as an explicit way to sample the likelihood of more fine-grained prompts (Equation 1) using a pre-trained language model (Betker et al., 2023), which effectively increases the diversity. It is possible to obtain higher diversity than the one captured by the real data when the prompt complexity is low, but this comes at the cost of consistency (as indicated in Figure 4). In CC12M, the diversity gap between vanilla guidance and real data slightly decreases with the increase of prompt complexity, indicating that the diversity for more general prompts is harder

---

[4]Unless otherwise specified, we use the DINOv2 (Oquab et al., 2023) feature space to compute these metrics, since it has been found to correlate well with human judgement (Hall et al., 2024a; Stein et al., 2023).

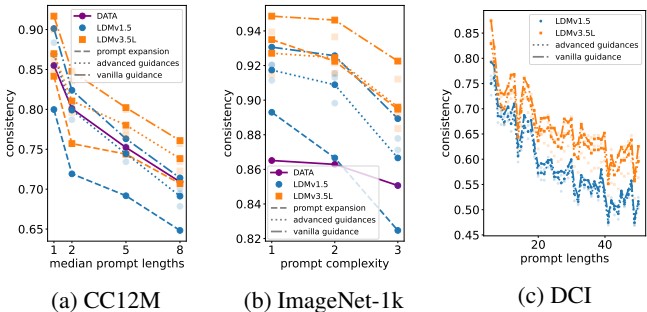

(a) CC12M          (b) ImageNet-1k          (c) DCI

Figure 4: **Reference-free consistency metric.** Consistency (DSG) metrics of LDMv1.5 and LDMv3.5L generations with CC12M and ImageNet-1k prompts when using: 1) vanilla guidance (CFG), 2) prompt expansion, and 3) advanced guidance methods, for which transparent markers correspond to different methods and the solid marker is the average over methods. Both advanced guidance methods and prompt expansion lead to lower consistency scores compared to vanilla guidance. We further extend to much longer DCI prompts. Consistency of all models decreases when the prompt lengths increases, which is the same as in the shorter prompt ranges.

to be captured, aligning with our synthetic experimental results (Figure 1b). However, ImageNet-1k exhibits a different pattern. This may be due to the construction of the real data prompt-image pairs, as general prompts can cover more subcategories than the ones in ImageNet-1k. For example, the general prompt (complexity=1) `stringed instrument` can contain mandolin (covered by T2I models as shown in Figure 20, Appendix H), but this class is not part of the ImageNet-1k dataset, leading to an underestimation of the real data diversity. We further push this evaluation towards much longer prompts using DCI dataset. We observe that the diversity decreases and then plateaus as we continuously increase the prompt length, indicating that the models may not be able to follow all the constraints added in the prompts (also reflected in the consistency measurements in Figure 4c). For most LDM models, the plateau starts at a prompt length of $\sim 30$ words. However, the plateau region of LDMv1.5 seems to occur earlier than its successor models.

**Quality.** Results across datasets and inference-time intervention methods are shown in Figure 3. In CC12M and ImageNet-1k datasets, where we use relatively shorter prompts, the aesthetics of the synthetic data remains relatively stable across prompt complexities, especially for the most recent model (LDMv3.5L). Further, advanced guidance methods lead to slightly lower performance than vanilla guidance. This is by contrast to prompt expansion, which consistently exhibits higher aesthetics than vanilla guidance. Compared to real data, synthetic data exhibits higher or competitive aesthetic quality. This is perhaps unsurprising given the aesthetics finetuning performed on the most recent models (LDMv3.5L). When we evaluate on the DCI dataset using much longer prompts, image aesthetics first increase and then start to gradually decrease as a function of prompt length for all models. The sharp slope observed towards more general prompts and the gradual decrease observed towards more fine-grained prompts appear aligned with the observation of our synthetic experiments that generalization to longer and fine-grained (higher complexity) prompts is easier than generalization to more general (lower complexity) ones, especially when leveraging CFG.

**Consistency.** Figure 4 shows the results across datasets and inference-time intervention methods. We observe that the prompt consistency decreases as we increase prompt complexity for all the cases considered. This suggests that T2I models struggle to incorporate the increasing amount of details (objects, attributes, and relations among objects) required by longer prompts, and to faithfully generate very specific concepts. Real data also shows a decrease in consistency across complexities, which is due to the image-text pairing process. The experiments on DCI dataset further consolidate this observation. We present the 95% confidence interval of these metrics in Appendix G.3, confirming the statistical significance of the observed trends. The findings connecting different utility axes and prompt complexity are visually captured in Figures 19 and 20 in Appendix H.

### 4.3 UTILITY AXES MEASURED BY REFERENCE-BASED METRICS

Reference-free metrics are suitable for evaluating the utility of synthetic data in the wild with prompts only. Yet, in our analysis, the prompts are extracted from existing datasets, *containing*

*both images and texts*, therefore enabling the use of reference-based metrics for evaluation too. The CC12M results are presented in Figures 5. We leave the ImageNet-1k results in Figure 13 in Appendix G.1. These figures reveal that as prompt complexity increases, there is an overall tendency for synthetic data to improve its precision, density and coverage in both CC12M and ImageNet-1k datasets, suggesting that more detailed captions, grounded on the real images, help generate data that lies within the support of the reference dataset. This also aligns with our synthetic experimental results showing that generalizing to general (lower complexity) prompts results in mode concentration and larger distributional gap (Figure 1b).

Figures 5 and 13 also show that advanced guidance methods and prompt expansion lead to overall better FDD than vanilla guidance. When it comes to coverage, prompt expansion also exhibits high performance compared to the vanilla guidance in most cases, although its benefits are less pronounced for higher complexity prompts. The improvements in coverage come, in both cases, at the expense of precision and density. This is perhaps expected as 1) prompt expansion may include details via pre-trained language models that are not present in the real images, therefore deviating the generation process from the reference data[5]; and 2) advanced guidance approaches may push the generation process outside of the real data manifold, resulting in reduced precision *w.r.t.* the vanilla guidance. The drop in precision and density shows the trade-off of creativity and fidelity to the real data distribution, emphasizing a cautious usage of T2I models for different downstream applications. We give a discussion in Appendix G.2.

Interestingly, LDMv1.5 model has better overall performance (lower FDD) compared to LDMv3.5L model across different guidance methods, prompt expansion, and prompt lengths. However, as shown in section 4.2, LDMv3.5L performs better than LDMv1.5 on reference-free quality and consistency metrics, only falling short in diversity, as shown in Figures 2a, 3a, and 4a. This suggests that the diversity is a key characteristic of real-world image distributions. Thus, synthetic data from models that fail to capture the diverse nature of the real world may lead to lower general performance in real-world applications even though the models advance the state-of-the-art generation quality in reference-free settings. Similar findings may be observed for the ImageNet-1k dataset, presented in Figures 2b, 3b, 4b, and 13.

### 4.4 COMBINING PROMPT EXPANSION AND ADVANCED GUIDANCE METHODS

Figure 6 shows the effect of combining prompt expansion and advanced guidance methods on the 3 utility axes of synthetic data, when using prompts from CC12M. We observe that the prompt expansion and advanced guidance methods can be combined to further boost the diversity of synthetic images. We note that in this case, Interval guidance appears to slightly suffer from aesthetic quality and shows considerably lower prompt consistency than both CFG and APG for prompts lengths higher than 1. Finally, when contrasting with real data, we observe that the diversity of the real data may be surpassed by that of synthetic data across prompt complexities, but this again comes at the cost of consistency. A closer look at advanced guidance methods and prompt expansion is provided in Appendix G.4.

## 5 RELATED WORK

**Improving the synthetic images utility.** Recent studies have yielded in-depth analyses on the utility of synthetic data from off-the-shelf T2I models (Astolfi et al., 2024; Hall et al., 2024c; Lee et al., 2023), and have revealed that the impressive progress in image quality has come at the expense of generation diversity. Inference-time interventions have been introduced to improve over classifier free guidance (CFG) sampling (Ho & Salimans, 2022). These techniques involve either prompt rewriting (Datta et al., 2024), or alternative guidance signals. For example, APG (Sadat et al., 2025) proposes to adaptively adjust the guidance scale across different directions, CADS (Sadat et al., 2024) implements an annealing strategy that gradually uses less noisier guidance signal during the sampling process, and Interval-guidance (Kynkäänniemi et al., 2024) introduces a simpler approach by disabling the conditional guidance at the beginning of the sampling process. Auto-guidance (Karras et al., 2024) uses a bad-version of the model to guide the generations;

---

[5]This can be understood as a mis-alignment between the likelihood estimation from the pre-trained language model and from the real image dataset.

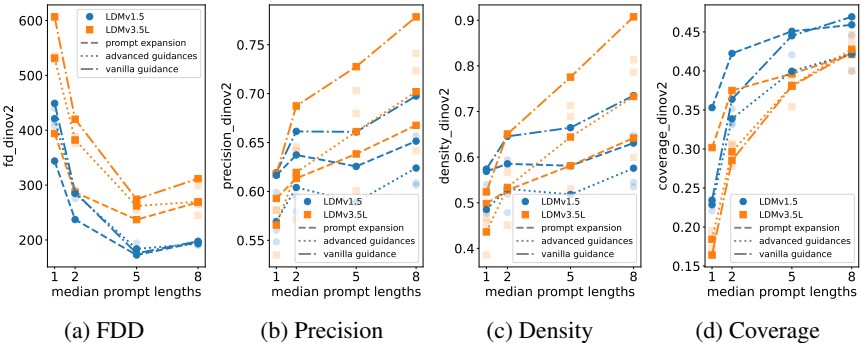

(a) FDD     (b) Precision     (c) Density     (d) Coverage

Figure 5: **Reference-based utility metrics of synthetic data using CC12M prompts.** FDD, precision, density and coverage for LDMv1.5 and LDMv3.5L generations with: (1) vanilla guidance (CFG), (2) prompt expansion, and (3) advanced guidance methods, for which transparent markers correspond to different methods and the solid marker is the average over methods. Both advanced guidance methods and prompt expansion lead to better FDD. Although prompt expansion improves coverage and advanced guidance methods match coverage for LDMv3.5, they both sacrifice precision and density. LDMv1.5 has thus better overall performance (lower FDD) than LDMv3.5L.

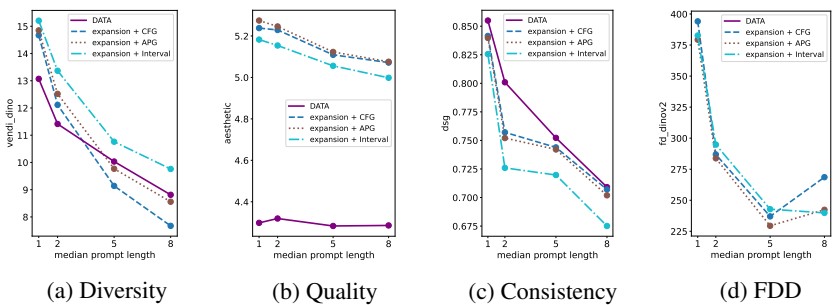

(a) Diversity     (b) Quality     (c) Consistency     (d) FDD

Figure 6: **Effect of combining prompt expansion and guidance methods on the utility of synthetic data from LDMv3.5L using CC12M prompts.** This can further boost the diversity of synthetic images, with comparable quality, consistency, and FDD (expecially with APG).

however, in the context of open-source models, it is not trivial how to obtain bad-versions of the models that would provide useful guidance signals. Chamfer guidance (Dall'Asen et al., 2025) leverages few examples of real images to guide the sampling process towards high-utility samples.

**Evaluation of T2I synthetic data.** Numerous evaluation metrics have been proposed to assess the performance of T2I models, covering the quality, diversity, consistency utility axes. Many metrics need a reference set of real data to compare with. For example, Fréchet Inception Distance (FID) (Heusel et al., 2017) has been widely used as an overall metric and measures the distance between the feature distributions of synthetic and real images. Precision and recall (Kynkäänniemi et al., 2019), followed by density and coverage (Naeem et al., 2020) were proposed to evaluate the realism and diversity of synthetic images. To evaluate T2I models in the wild, researchers have introduced several reference-free metrics. When it comes to quality, aesthetics score (Schuhmann, 2022; discus0434 & Goswami, 2023) has been built on top of CLIP (Radford et al., 2021) to estimate how visually appealing synthetic images are. For diversity, Vendi score (Friedman & Dieng, 2023) was designed to evaluate the diversity of a set of images. Moreover, in Tang et al. (2024), a human-calibrated perceptual variability measure was introduced and then used to investigate the influence of linguistic features in the T2I generation process. This work focuses on the diversity axes exclusively, does not contrast results with real data, and does not consider any sampling intervention beyond vanilla guidance. For prompt-image consistency, CLIPscore (Hessel et al., 2021) has been used to measure the alignment between the generated image and the input text prompt. However, CLIPscore has been shown to be correlated with object numerosity (Ahmadi & Agrawal, 2024) and unable to handle detailed descriptions (Zhang et al., 2024). VQAscore (Lin et al., 2024),

TIFA (Hu et al., 2023) and DSG (Cho et al., 2024) further leverage visual question answering (VQA) models for image-prompt consistency. Among them, TIFA and DSG decompose the captions into several granular questions and compute accuracy scores based on the answers to each question. Beyond these single-axis utility metrics, several analyses have been performed on T2I models from a multi-objective perspective (Astolfi et al., 2024; Lee et al., 2023; Hall et al., 2024b). Our study complements previous work by assessing the utility of synthetic data as a function on prompt complexity.

## 6    Conclusion

**Conclusion.**    In this paper, we proposed a new evaluation framework and presented an in-depth analysis of the utility of synthetic data as a function of prompt complexity, including synthetic and large-scale real data evaluations. Our synthetic experiments show that generalizing to more general conditions is harder than the other way round. Our study suggests that prompt complexity is a crucial axis to consider when sampling from T2I models, and requires more investigation when generating from very general prompts. Our findings include 1) the trend of utilities are non-linear, showing an asymmetry of prompt length generalization, 2) diversity does not collapse but plateaus as prompt length increases, suggesting an inherent "lower bound of diversity" in T2I models, 3) optimizing for reference-free metrics harms distributional fidelity, 4) combining guidance methods with prompt expansion achieve the most interesting trade-offs in utilities. In addition, diversity is a key feature of real-world image distributions, which is still not properly captured by synthetic images from the state-of-the-art T2I models when no explicit prompt expansion is conducted during inference.

**Limitations.**    For datasets with long captions, our framework needs to use large and diverse enough image datasets to build paired image sets, which is not easy to get. Since DCI dataset has only 7805 image-caption pairs, we did not compare the utilities of synthetic images with real data.

## Reproducibility statement

We detailed our experimental setups, including datasets, hyperparameters, training and inference settings, and computing resources, in both main texts (Section 2 and 4.1) and Appendices (Appendix A.1 and C). We provided full derivations of our theoretical results in Appendix A.2. All the datasets used in our paper are open-source and can be downloaded following the instructions given by dataset providers. We detailed the licences of datasets and packages used in Appendix C.5. We provided detailed descriptions of our evaluation framework in Section 3 and Appendix B, and our anonymous codes in supplementary materials for reviewing.

## Acknowledgment

AC's work was supported by the Institute of Information & Communications Technology Planning & Evaluation (IITP) grant funded by the Korean Government (MSIT) (No. RS-2024-00457882, National AI Research Lab Project) and his own CIFAR Canadian AI Chair. XZ is supported by the AIM program at Meta and the EDI in Research Scholarship of Mila. The authors thank Oscar Mañas, Andrei Nicolicioiu, and Nicola Dall'Asen for insightful discussions regarding the evaluation pipeline; Avery HW Ryoo for helpful suggestions on prompting language models; and Nicolas Beltran-Velez and Felix Friedrich for constructive feedback during the rebuttal.

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

# Appendix

# A  DETAILS ON THE SYNTHETIC EXPERIMENTS

## A.1  EXPERIMENTAL SETTINGS

We use a mixture of four Gaussians as our training distribution. Each Gaussian is in each quadrant respectively. The means for each Gaussian is [-3, 3], [-3, -3], [3, 3], and [3, -3]. The covariance for each Gaussian is [[0.7,0],[0,0.7]]. We hypothetically link each Gaussian to a conditional prompt, *i.e.*, `white cat` for the first quadrant, `white dog` for the second quadrant, `black dog` for the third quadrant and `black cat` for the fourth quadrant.

We train conditional U-Net-based diffusion model with DDPM schedule using two types of conditional prompts: one is fine-grained with `white cat`, `white dog`, `black cat`, `black dog`, each corresponds to a Gaussian in one quadrant; the other is general with `cat`, `dog`, `white`, and `black`, each corresponds to two Gaussians.

Following Rombach et al. (2022); Esser et al. (2024), we add a pooled conditional information together with the timestep embedding, and also incorporate the token-wise conditional information into the U-Net with an Attention mechanism. To get the pooled conditional information, we use a linear layer, while to get the token-wise embedding, we initialize a vocabulary dictionary from scratch and train it together with the diffusion model. We train the model with batch size 512, Adam optimizer, learning rate 1e-4 with linear warmup and exponential decay ($\gamma = 0.99$), training epochs 250 in total. All training and inference are done using single NVIDIA V100 32GB GPU.

During training, we adopt a 50% probability of dropping the conditional information, enabling the classifier-free guidance (CFG) (Ho & Salimans, 2022) during inference. We use the ancestral sampler in DDPM (Ho et al., 2020) with 1000 steps for inference. We conduct the inference in a generalization setting, *i.e.*, if the model is trained with general prompts, the inference is done with fine-grained prompts, and vice versa. We use forward Kullback-Leibler divergence and Fréchet Distance to evaluate the generated samples. We sample 10,000 data points for evaluation.

## A.2  DERIVATIONS

**From general conditions to fine-grained conditions.** Given a conditional diffusion model with parameters $\theta$ trained on general conditions, timestep $t \in \{1, 2, ..., T\}$, independent general conditions $c_g^i, i \in \mathbb{N}$, we have access to their score functions $s_\theta(x_t|c_g^i) = \nabla_{x_t} \log p_\theta(x_t|c_g^i)$. Supposing that a fine-grained condition $c_f$ comprises of several general conditions $c_g^i, i \in \{1, 2, ..., K\}$, we have

$$p_\theta(x_t|c_f) = p_\theta(x_t| \cap_{i\in\{1,2,...,K\}} c_g^i) \tag{3}$$

$$= \frac{p_\theta(x_t, c_g^1, c_g^2, ..., c_g^K)}{p_\theta(c_g^1, c_g^2, ..., c_g^K)} \tag{4}$$

$$= \frac{p_\theta(x_t) \prod_{i\in\{1,2,...,K\}} p_\theta(c_g^i|x_t)}{\prod_{i\in\{1,2,...,K\}} p_\theta(c_g^i)} \tag{5}$$

$$= \frac{p_\theta(x_t) \prod_{i\in\{1,2,...,K\}} \frac{p_\theta(x_t|c_g^i)p_\theta(c_g^i)}{p_\theta(x_t)}}{\prod_{i\in\{1,2,...,K\}} p_\theta(c_g^i)} \tag{6}$$

$$= p_\theta(x_t) \prod_{i\in\{1,2,...,K\}} \frac{p_\theta(x_t|c_g^i)}{p_\theta(x_t)} \tag{7}$$

$$s_\theta(x_t|c_f) = \nabla_{x_t} \log p_\theta(x_t|c_f) \tag{8}$$

$$= \nabla_{x_t} \log p_\theta(x_t) + \nabla_{x_t} \sum_{i\in\{1,2,...,K\}} \left( \log p_\theta(x_t|c_g^i) - \log p_\theta(x_t) \right) \tag{9}$$

$$= s_\theta(x_t) + \sum_{i\in\{1,2,...,K\}} \left( s_\theta(x_t|c_g^i) - s_\theta(x_t) \right) \tag{10}$$

**From fine-grained conditions to general conditions.** Given a conditional diffusion model with parameters $\theta$ trained on fine-grained conditions, timestep $t \in \{1, 2, ..., T\}$, independent fine-grained conditions $c_f^i, i \in \mathbb{N}$, we have access to their score functions $s_\theta(x_t | c_f^i) = \nabla_{x_t} \log p_\theta(x_t | c_f^i)$. Supposing that a general condition $c_g$ can be decomposed into different fine-grained conditions $c_f^i, i \in \{1, 2, ..., M\}$, we have

$$p_\theta(x_t | c_g) = \sum_{i \in \{1,2,...,M\}} p_\theta(x_t | c_f^i) \tag{11}$$

$$s_\theta(x_t | c_g) = \nabla_{x_t} \log \sum_{i \in \{1,2,...,M\}} p_\theta(x_t | c_f^i) \tag{12}$$

$$= \frac{1}{\sum_{j \in \{1,2,...,M\}} p_\theta(x_t | c_f^j)} \nabla_{x_t} \left( \sum_{i \in \{1,2,...,M\}} p_\theta(x_t | c_f^i) \right) \tag{13}$$

$$= \frac{1}{\sum_{j \in \{1,2,...,M\}} p_\theta(x_t | c_f^j)} \sum_{i \in \{1,2,...,M\}} \nabla_{x_t} p_\theta(x_t | c_f^i) \tag{14}$$

$$= \frac{1}{\sum_{j \in \{1,2,...,M\}} p_\theta(x_t | c_f^j)} \sum_{i \in \{1,2,...,M\}} \left( p_\theta(x_t | c_f^i) \nabla_{x_t} \log p_\theta(x_t | c_f^i) \right) \tag{15}$$

$$= \sum_{i \in \{1,2,...,M\}} \left( \frac{p_\theta(x_t | c_f^i)}{\sum_{j \in \{1,2,...,M\}} p_\theta(x_t | c_f^j)} s_\theta(x_t | c_f^i) \right) \tag{16}$$

### A.3 Relationship with the real T2I settings

We use the synthetic experiment as a motivation to emphasize the importance of studying the axis of prompt complexity in the diffusion / flow model setting. The synthetic experiments clearly show an asymmetry of difficulty w.r.t. conditioning complexity when the probabilistic assumptions hold.

The conditional independence assumption in synthetic setting derivation is a simplification. Real-world text encoders (*e.g.*, CLIP/T5) produce entangled representations where concepts (*e.g.*, "yellow" and "banana") can be highly correlated and content dependent. However, we believe this distinction does not invalidate our conclusion regarding the asymmetry of difficulty between generalizing to general ("OR") vs. specific ("AND") prompts.

#### A.3.1 On how this affects the "OR is harder than AND" statement

**AND Operator** (Generalizing to longer prompts):

In our toy example (Ideal Case): The "black" and "dog" concepts are independent. Their guidance vectors are orthogonal. Our Equation (2) is a probabilistically sound formula, and the naive sum works perfectly, as shown by the KL/FD scores (KL divergence of 0.93 and Frechet distance of 1.32).

The violation of the assumption doesn't make this operator "harder" in the sense of being impractical, but degrades it. It changes Equation (2) from a probabilistic law into a practical but imperfect heuristic, probably leading to some failure modes. For example, the guidance vectors for "white" and "dog" are non-orthogonal if they are correlated as captured by the CLIP / T5 model. Our Equation (2), by naively adding these vectors, leads to an over-magnification of this shared, correlated signal, potentially leading to some artifacts (low diversity, over-saturation, etc.) in the generation.

**OR Operator** (Generalizing to shorter prompts):

This is already hard in theory as we explain in Eq. 1 in our manuscript, where the score function for "OR" operator is intractable in diffusion models. The fact that tokens in real-world prompts are not perfectly "mutually exclusive" adds another layer of intractability, making this "OR" generalization even more difficult.

Given the theoretical analysis and not over-interpret our empirical results, some of our empirical results also verifies this asymmetry. For aesthetic quality with CC12M and DCI settings (Fig. 3, (a)

and (c)), we observe a sharper decrease towards shorter prompts compared to longer prompts, which empirically supports our statement.

### A.3.2 ON EXPLAINING CC12M VS. IMAGENET BEHAVIORS

Our theoretical analysis also helps explain some empirical divergences in results of CC12M and ImageNet. CC12M relies on composition of tokens (similar to the "AND" and "OR" operator logic), where we see similar trends as in the synthetic experiments. ImageNet experiments rely on specificity (using semantically richer tokens rather than more tokens) and the ImageNet hierarchy does not strictly follow the combinatorial logic of these mathematical derivations.

## B DETAILS ON THE BENCHMARKING FRAMEWORK

### B.1 CAPTIONING STEP

We use Gemma3 (Team et al., 2025) model (`gemma-3-12b-it` version) to conduct the captioning of images from datasets with general captions (*e.g.*, CC12M (Changpinyo et al., 2021)). The system prompt is: `You strictly follow the user's instructions.` The user prompts used for captioning to different complexities are as follows:

1. `Mention the main subjects in the image with only 1 word.  Do not use specific identifiers, such as names.  Format the caption as such:  <<CAPTION>>.`

2. `Write a short caption with 3 or less words.  Use only one adjective.  Do not use specific identifiers, such as names. Format the caption as such:  <<CAPTION>>.`

3. `Write a caption with 6 or less words, describing at most two main foreground subjects.  Do not use specific identifiers, such as names.  Format the caption as such:  <<CAPTION>>.`

4. `Write a caption with 8 or less words, describing at most two main foreground subjects, as well as the background.  Do not use specific identifiers, such as names.  Format the caption as such:  <<CAPTION>>.`

### B.2 PAIRING STEP

For datasets with general captions, we use the SigLip model (Zhai et al., 2023) to embed both images and texts. Our implementation uses the OpenClip [6] library and the SigLip version `ViT-SO400M-14-SigLIP-384`.

### B.3 ALIGNMENT STEP

We present the alignment step in Algorithm 1.

### B.4 EVALUATION METRICS

To compute evaluation metrics, we use the following libraries:

1. Reference-based metrics: `dgm-eval` (Stein et al., 2023)

2. Vendi score: `Vendi-Score` (Friedman & Dieng, 2023)

3. Aesthetic score: `aesthetic-predictor-v2.5` (discus0434 & Goswami, 2023)

4. DSG score: `Eval-GIM` (Hall et al., 2024b)

---

[6] https://github.com/mlfoundations/open_clip

**Algorithm 1** Alignment step

1: **Input:** $\bar{\mathcal{I}}^{ik}, k \in \{1, 2, ..., K\}, i \in \mathcal{N}_k^{\mathrm{p}}; N_{\mathrm{gen}}$.
2: breaksignal $\leftarrow$ `False`
3: **while** *not* breaksignal **do**
4:     CommonImagesBefore $\leftarrow \bigcap_{k \in \{1,2,...,K\}} \bigcup_{i \in \mathcal{N}_k^{\mathrm{p}}} \bar{\mathcal{I}}^{ik}$
5:     **for** $j \in \{1, 2, \ldots, K\}$ **do**
6:         **for** $l \in \mathcal{N}_j^{\mathrm{p}}$ **do**
7:             $\bar{\mathcal{I}}^{lj} \leftarrow \bar{\mathcal{I}}^{lj} \cap$ CommonImagesBefore
8:             **if** $|\bar{\mathcal{I}}^{lj}| < N_{\mathrm{gen}}$ **then**
9:                 **del** $l$ from $\mathcal{N}_j^{\mathrm{p}}$
10:             **end if**
11:         **end for**
12:     **end for**
13:     CommonImagesAfter $\leftarrow \bigcap_{k \in \{1,2,...,K\}} \bigcup_{i \in \mathcal{N}_k^{\mathrm{p}}} \bar{\mathcal{I}}^{ik}$
14:     breaksignal $\leftarrow$ len(CommonImagesBefore) == len(CommonImagesAfter)
15: **end while**
16: **return** $\bar{\mathcal{I}}^{ik}, k \in \{1, 2, ..., K\}, i \in \mathcal{N}_k^{\mathrm{p}}$

## C  EXPERIMENTAL DETAILS

### C.1  DATASET DETAILS

*CC12M.* We investigate the effect of increasing prompt detail and length on the utility axes of synthetic data. Following the procedure described in Section 3, we generate captions for randomly sampled one million images at four different complexity levels, each corresponding to a distinct complexity level. Lower complexity prompts are shorter and contain fewer details about the image. We follow the procedure in Section 3 and sample 5,000 captions per complexity for generation. As reference, many widely used evaluation prompt sets contain approximately 1,000 prompts (Cho et al., 2024; Hu et al., 2024). We produce 20 images per prompt, resulting in 100,000 generated images for each prompt complexity. The statistics for different caption complexities are presented in Table 1. We use `c#` to represent complexity level of the prompts. We report the caption lengths statistics across complexities, and the number of real images retrieved using the 5,000 selected captions per complexity.

Table 1: Statistics on word lengths of different prompt complexities

|  | c1 | c2 | c3 | c4 |
|---|---|---|---|---|
| Avg # word | 1.0007 | 2.1539 | 4.9094 | 7.5410 |
| Std # word | 0.0501 | 0.3621 | 0.7032 | 0.8849 |
| Median # word | 1.0000 | 2.0000 | 5.0000 | 8.0000 |
| # real images covered | 61,334 | 57,925 | 55,075 | 46,066 |

*ImageNet-1k.* We examine the effect of concept specificity on the utility axes of synthetic data. We follow the procedure in Section 3 and extract hyponym relations to vary the specificity of ImageNet-1k class labels. For example, the class `siberian husky` (highest specificity) has the parent class `sled dog` and grandparent class `working dog` (lowest specificity). Since these class labels possess intrinsic semantic complexity, we use class-label specificity as a proxy for complexity. To generate images, we use the prompt "Image of a `<CLASS>`". We utilize all 1,000 ImageNet-1k classes and generate 50 images per class, resulting in 50,000 images, matching the size of the ImageNet-1k evaluation set.

*DCI.* We further explore prompt complexity using *very long* prompts. DCI's detailed captions consist of multiple sentences, densely describing various aspects of each image. We construct prompts as described in follow the procedure in Section 3, *i.e.*, progressively concatenating sentences to each

caption. We cap the maximum prompt length at 100 words. We sample 1,000 detailed captions and generate 5,398 prompts of varying complexity. For each prompt, we generate 50 images.

## C.2 GUIDANCE METHODS' HYPERPARAMETERS

For inference, we use Euler discrete sampling for all guidance methods and sample 28 steps. The timestep scale for all models is from 0 to 1. To find hyperparamers for our setting, we start from the original papers' setting (when the information is available for the models used in our paper) and then empirically test the generation based on a randomly selected set of 10 prompts. Since many methods are tested on class-conditional models, the hyperparameters are not always directly applicable to our setting. Empirically, we find that we should keep more conditional information to achieve enough image-prompt consistency compared to class-conditional models.

*APG (Sadat et al., 2025).* This guidance method has three hyperparameters: effect of the parallel component $\eta$, effect of the rescaling threshold $r$, and effect of the momentum strength $\beta$. We follow the original paper and set $\eta = 0$ for all settings. Other hyperparameters are presented in Table 2.

Table 2: Hyperparameters for APG guidance method.

| params | LDMv1.5 | LDM-XL | LDMv3.5M | LDMv3.5L |
|:---:|:---:|:---:|:---:|:---:|
| $r$ | 7.5 | 15 | 10 | 10 |
| $\beta$ | -0.75 | -0.5 | -0.5 | -0.5 |

*Interval (Kynkäänniemi et al., 2024).* This guidance method has two hyperparameters: $\tau_{\mathrm{lo}}$ and $\tau_{\mathrm{hi}}$. The conditional information is only applied between $\tau_{\mathrm{lo}}$ and $\tau_{\mathrm{hi}}$. The hyperparameters used in our analyses are presented in Table 3.

Table 3: Hyperparameters for interval guidance method.

| params | LDMv1.5 | LDM-XL | LDMv3.5M | LDMv3.5L |
|:---:|:---:|:---:|:---:|:---:|
| $\tau_{lo}$ | 0.08 | 0.08 | 0.3 | 0.3 |
| $\tau_{hi}$ | 0.81 | 0.81 | 0.95 | 0.95 |

*CADS (Sadat et al., 2024).* This guidance method has two main hyperparameters to control how the conditional information is annealed: $\tau_1$ and $\tau_2$, where the guidance scale is set to 0 for time steps between $\tau_2$ and 1.0, then linearly increases between time steps $\tau_1$ and $\tau_2$ from 0.0 to 1.0, and is kept to 1.0 between time steps 0.0 and $\tau_1$. In addition, CADS also has hyperparameters for noise scale $s$ and for mixing factor $\phi$. We keep $\phi = 1$ for all settings following the original paper. We present the hyperparameters' choice of CADS for different models in Table 4. Some of the $\tau_2$ values are larger than 1.0, which means that we always keep some conditional information during sampling. This can help achieve better image-prompt consistency.

Table 4: Hyperparameters for CADS guidance method.

| params | LDMv1.5 | LDM-XL | LDMv3.5M | LDMv3.5L |
|:---:|:---:|:---:|:---:|:---:|
| $\tau_1$ | 0.8 | 0.6 | 0.85 | 0.85 |
| $\tau_2$ | 1.3 | 1.0 | 1.25 | 1.25 |
| $s$ | 0.1 | 0.3 | 0.3 | 0.3 |

## C.3 PROMPT EXPANSION

We use the Gemma3 model (Team et al., 2025) to expand the prompts. We use the `gemma-3-12b-it` version. The prompt used for expansion is as follows:

Table 5: Licences for all the models, datasets and libraries used in our paper.

| Asset | Type | Licence |
|---|---|---|
| CC12M (Changpinyo et al., 2021) | Dataset | Freely use for any purpose [7] |
| Imagenet-1k (Deng et al., 2009) | Dataset | Custom non-commercial license |
| DCI (Urbanek et al., 2024) | Dataset | CC BY-NC 4.0 and SA-1B dataset Licnese [8] |
| LDMv3.5L (Esser et al., 2024) | Model | Stability Community License |
| LDMv3.5M (Esser et al., 2024) | Model | Stability Community License |
| LDM-XL (Podell et al., 2024) | Model | CreativeML Open RAIL++-M License |
| LDMv1.5 (Rombach et al., 2022) | Model | CreativeML Open RAIL-M License |
| Gemma3 (Team et al., 2025) | Model | Gemma License |
| SigLip (Zhai et al., 2023) | Model | Apache-2.0 |
| Openclip [9] | Library | Openclip License [10] |
| Vendi-Score (Friedman & Dieng, 2023) | Library | MIT License |
| DGM-eval (Stein et al., 2023) | Library | MIT License |
| Eval-GIM (Hall et al., 2024b) | Library | CC BY-NC 4.0 |
| Aesthetic-predictor-v2.5 (discus0434 & Goswami, 2023) | Library | AGPL-3.0 |

- system prompt: `The user will give a sentence that is a caption for an image. Based on the sentence, you will generate 20 different new sentences with around 30 words, that don't violate the original meaning, but with more possible details in that image. You will return the sentences in a list format.`

- user prompt: `The sentence given is: ``CAPTION''.`

### C.4 COMPUTE RESOURCES

We use our internal cluster to conduct all the experiments. The captioning, pairing and alignment steps are conducted on NVIDIA V100 GPUs with 32GB memory. The image generation is conducted on NVIDIA H100 GPUs with 80GB memory. All the models are run in 16-bit precision and can fit into one GPU. The generation time for 100,000 images is around 175 GPU hours for LDMv3.5L (Esser et al., 2024), 80 GPU hours for LDMv3.5M (Esser et al., 2024), 40 GPU hours for LDM-XL (Podell et al., 2024) and 4 GPU hours for LDMv1.5.

### C.5 LICENSES FOR EXISTING ASSETS

We list licenses for all the models, datasets and libraries used in our paper in Table 5.

### C.6 NEW ASSETS

Our code is under License CC BY-NC 4.0.

## D RESULTS FOR LDM-XL AND LDMv3.5M

In this section, we present results for LDM-XL (Podell et al., 2024) and LDMv3.5M (Esser et al., 2024) models on CC12M dataset (Changpinyo et al., 2021) and ImageNet-1k dataset (Deng et al., 2009), complementing the results in the main paper.

---

[7] https://github.com/google-research-datasets/conceptual-12m?tab=License-1-ov-file#readme

[8] https://ai.meta.com/datasets/segment-anything-downloads/

[9] https://github.com/mlfoundations/open_clip

[10] https://github.com/mlfoundations/open_clip?tab=License-1-ov-file#readme

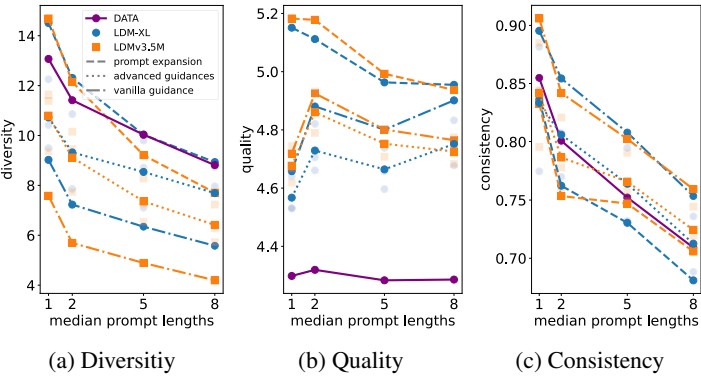

(a) Diversitiy     (b) Quality     (c) Consistency

Figure 7: **Reference-free utility metrics of synthetic data using CC12M prompts.** Diversity (Vendi), quality (aesthetics) and consistency (DSG) metrics of LDM-XL and LDMv3.5M generations when using: 1) vanilla guidance (CFG), 2) prompt expansion, and 3) advanced guidance methods, for which transparent markers correspond to different methods and the solid marker is the average over methods. Both advanced guidance methods and prompt expansion lead to improved diversity over the vanilla guidance. Prompt expansion from shorter captions can surpass the real data diversity. Prompt expansion leads to quality improvements, whereas advanced guidance methods slightly reduces quality *w.r.t.* vanilla guidance.

## D.1 UTILITY AXES MEASURED BY REFERENCE-FREE METRICS

We measure the utility of synthetic data in terms of quality (aesthetics), diversity (Vendi) and prompt consistency (DSG) with reference-free metrics. We present results using the prompts from CC12M in Figure 7 and ImageNet-1k in Figure 8. The results show the same trend as presented in Section 4.2.

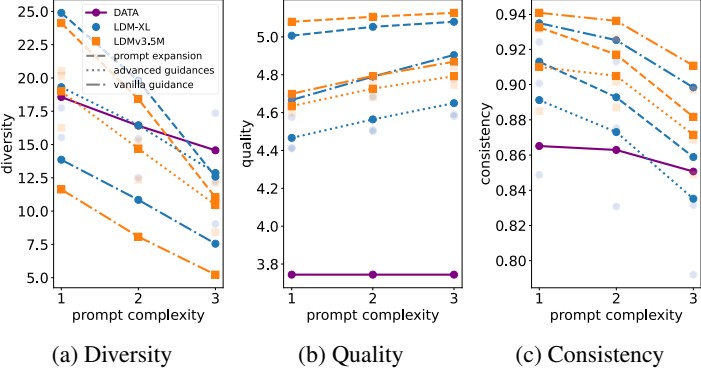

(a) Diversity     (b) Quality     (c) Consistency

Figure 8: **Reference-free utility metrics of synthetic data using ImageNet-1k class labels.** Diversity (Vendi), quality (aesthetics) and consistency (DSG) metrics for LDM-XL and LDMv3.5M generations with: 1) vanilla guidance (CFG), 2) prompt expansion, and 3) advanced guidance methods, for which transparent markers correspond to different methods and the solid marker is the average over methods. Prompt complexity shows the specificity of the class label used for generation. Both advanced guidance methods and prompt expansion lead to higher diversity, but advanced guidance methods lead to slightly lower quality. Prompt expansion and advanced guidances with shorter captions can yield higher diversity compared to the real data. Both prompt expansion and advanced guidances have a negative impact on consistency.

## D.2 UTILITY AXES MEASURED BY REFERENCE-BASED METRICS

Reference-free metrics are suitable for evaluating the utility of synthetic data in the wild. In our analysis, the prompts are extracted from existing dataset, containing both images and texts, and

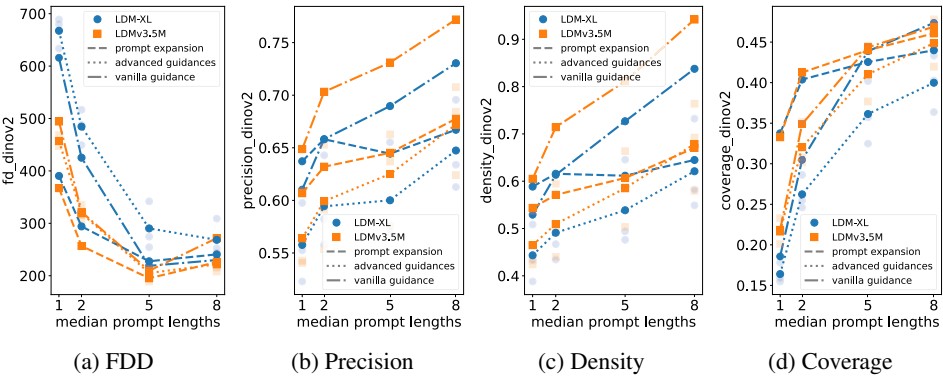

(a) FDD  (b) Precision  (c) Density  (d) Coverage

Figure 9: **Reference-based utility metrics of synthetic data using CC12M prompts.** Fréchet distance (FDD), precision, density and coverage in the DINOv2 feature space for LDM-XL and LDMv3.5M generations with: (1) vanilla guidance (CFG), (2) prompt expansion, and (3) advanced guidance methods, for which transparent markers correspond to different methods and the solid marker is the average over methods. Prompt expansion lead to better FDD for both models. Both prompt expansion and advanced guidances sacrifice precision and density.

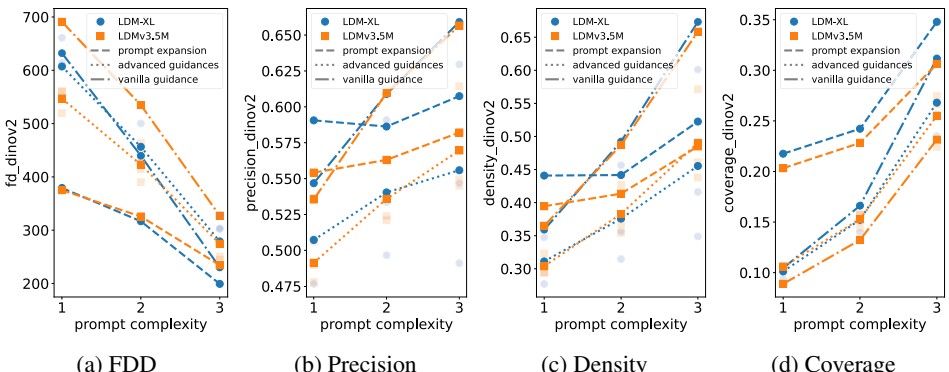

(a) FDD  (b) Precision  (c) Density  (d) Coverage

Figure 10: **Reference-based utility metrics of synthetic data using ImageNet-1k class labels.** FDD, precision, density, and coverage metrics in the DINOv2 feature space for LDM-XL and LDMv3.5M with: 1) vanilla guidance (CFG), 2) prompt expansion, and 3) advanced guidance methods, for which transparent markers correspond to different methods and the solid marker is the average over methods. Both advanced guidance methods and prompt expansion lead to better FDD, and improve or match the coverage of vanilla guidance, while sacrificing precision and density in most cases. For LDMv3.5M, we see a more prominent effect of advanced guidance methods and prompt expansion than for LDM-XL. Advanced guidance methods do not change the trend of the metrics *w.r.t.* prompt complexity, while the prompt expansion shapes the trend of the metrics differently.

so enabling the use of reference-based metrics for evaluation. Unless otherwise specified, we use the DINOv2 (Oquab et al., 2023) feature space to compute these metrics, since it has been found to correlate well with human judgement (Hall et al., 2024a; Stein et al., 2023). The CC12M and ImageNet-1k results are presented in Figures 9 and 10, respectively.

The results are similar to those for LDMv1.5 and LDMv3.5L presented in Section 4.3. It is worth noting that we employed more aggressive hyperparameters of advanced guidance methods for LDM-XL model as shown in Tables 2, 3 and 4. This leads to larger differences for reference-based metrics since the generated images can be out side of the support of the reference dataset.

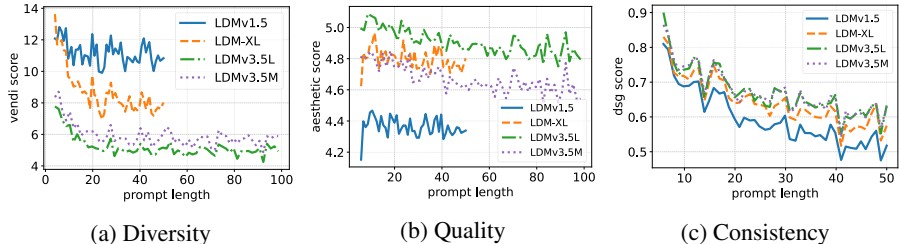

| (a) Diversity | (b) Quality | (c) Consistency |

Figure 11: **Reference-free utility metrics of synthetic data using DCI prompts of increasing complexity.** For all models, diversity decreases at first and then plateaus, while consistency score continuously decreases with the increase of the prompt length. The quality of LDMv3.5 decreases as prompt complexity increases.

### D.3 DCI RESULTS

DCI dataset opens the possibility to evaluate the utility of synthetic data for notably longer prompts. We perform reference-free evaluations on synthetic data generated from DCI prompts by binning the prompts into different length chunks and computing the average diversity, aesthetic and consistency scores for each bin. We present the results in Figure 11 [11]. We observe that the diversity decreases and then plateaus as we increase the prompt length. For most LDM models, the plateau starts at a prompt length of $\sim 30$ words. However, the plateau region of LDMv1.5 seems to occur earlier than for its successor models. However, the consistency tends to decrease as we increase the prompt length. As the prompt becomes increasing detailed, the consistency metric keeps assessing whether all the information is captured, emphasizing the limitations of current T2I models to follow very long and detailed prompts. Interestingly, for the most recent LDMv3.5 models, image aesthetics also seems to slightly decrease as a function of prompt length.

## E HUMAN EVALUATION

Grounding our automatic metrics with human evaluation is crucial for validating our findings. We conducted human evaluations, specifically designed to assess whether our automatic diversity metric (Vendi score) and consistency metric (DSG score) aligns with human perception.

### E.1 VENDI SCORE

We employed a two-alternative forced-choice (2AFC) task. In each trial, annotators were shown two image mosaics (16 images each), generated from prompts of two different complexity levels. They were asked: "Which set of images is more diverse?" Images were generated from CC12M grounded prompts of different complexities using the LDMv3.5L model. For each trial, we randomly sampled two prompts, each from a distinct complexity level to generate the two mosaics for comparison. The study involved 9 annotators, who collectively provided 900 pairwise comparison annotations, ensuring comprehensive coverage of all complexity level pairs. We calculated the win rate for lower-complexity prompts against higher-complexity prompts. A win rate greater than 0.50 indicates that, on average, humans perceive images from lower-complexity prompts as more diverse. The results are summarized in Table 6.

Our analysis yields two key findings that validate our approach:

1. **Human perception aligns with our main findings.** In all pairwise comparisons, the win rate for the lower-complexity prompts was all above 0.50 (e.g., 0.620 for complexity 1 vs. 3; 0.794 for 0 vs. 3). This proves that humans consistently perceive the sets of images generated from simpler prompts as more diverse, mirroring the trend identified by our automatic metrics.

2. **Vendi score strongly correlates with human preference.** There is a clear positive correlation between the magnitude of the Vendi score difference (values in parentheses) and the human-annotated win rate. For instance, the largest Vendi score difference (3.00, between

---

[11]LDMv1.5 and LDM-XL models stop at prompt length of 50, given the 77 token limit of their text encoder.

levels 0 and 3) corresponds to the most decisive human judgment (0.794 win rate), while the smallest difference (0.605, between levels 2 and 3) corresponds to a judgment barely above chance (0.531 win rate). This provides strong evidence that the Vendi score used in our work is a meaningful and valid proxy for the degree of human-perceived diversity.

Table 6: Human evaluation results for diversity. Values represent the win rate of images generated from prompts of the row complexity level being judged as more diverse than those from the column level. Values in parentheses show the corresponding Vendi score difference between the two levels.

| Complexity level | 2 | 3 | 4 |
|---|---|---|---|
| 1 | 0.589 (1.63) | 0.645 (2.40) | 0.794 (3.00) |
| 2 | - | 0.563 (0.77) | 0.620 (1.37) |
| 3 | - | - | 0.531 (0.61) |

### E.2 DSG SCORE

DSG uses a two-stage process: (1) Decomposition: An LLM (Gemma-3-27B-Instruct, the largest model in the series released in 2025) decomposes the single complex prompt (e.g., "A white dog is playing on the grassy field.") into a set of simple, atomic questions (e.g., "Is there a dog?", "Is the dog white?", "Is the field grassy?", ... ). (2) VQA: The VQA model (default as in DSG score paper) is then only required to answer these simple, unit-level yes/no questions, a task it can perform more reliably compared to answering a single long complex question.

To further validate our automatic evaluated results with DSG score, we conduct a human evaluation for both stages.

We give the guideline on how the prompt is decomposed into atomic questions along with an example of decomposition presented to human annotators. Given a prompt, we present the prompt along with the decomposed questions generated by LLM to human evaluators, and ask them to check whether each question is a valid question according to decomposition rules. We ask 4 human annotators, and each of them annotates 25 prompts, yielding 100 sets of answers in total. Among all the decomposed questions, the percentage of correctly decomposed questions is 95.54%, which effectively demonstrates the validity of the decomposed questions using LLM automatic generation.

Given the generated images from the original long prompt, we ask the human annotators to answer each decomposed question generated by LLM from the original long prompt. We ask 10 annotators and each evaluates 50 generated images using DCI prompts and gather in total 500 evaluation results. The human evaluation is small compared to the large-scale automatic evaluation, but it shows the same decreasing trend and a strong correlation between human evaluation results and automatic VLM results. Table 7 shows the human evaluation results for Stage 2.

Table 7: DSG metrics computed using human evaluation and automatic VLM

| # words | 10 | 13 | 18 | 23 | 28 | 35 | 41 | 50 |
|---|---|---|---|---|---|---|---|---|
| # samples | 58 | 54 | 66 | 61 | 65 | 66 | 65 | 75 |
| Human DSG | 0.92 | 0.90 | 0.85 | 0.87 | 0.83 | 0.82 | 0.81 | 0.81 |
| VQA DSG | 0.95 | 0.89 | 0.81 | 0.86 | 0.85 | 0.81 | 0.79 | 0.82 |
| Pearson corr 0.89; R2 0.68 | | | | | | | | |

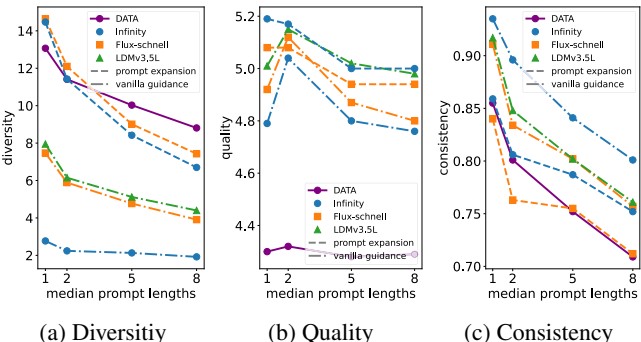

(a) Diversitiy      (b) Quality      (c) Consistency

Figure 12: **Reference-free utility metrics of synthetic data using CC12M prompts.** Diversity (Vendi), quality (aesthetics) and consistency (DSG) metrics of Infinity and Flux-schnell generations when using: 1) vanilla guidance (CFG) and 2) prompt expansion. Prompt expansion leads to improved diversity over the vanilla guidance. Prompt expansion from shorter captions can surpass the real data diversity. Prompt expansion leads to quality improvements. Infinity model with vanilla guidance has very low diversity in generations, while prompt expansion can effectively recover the diversity.

## F  ADDITIONAL MODELS

Our selection of models cover latent diffusion models and flow models with a chronological perspective. We can compare different guidance methods over all these models selected. However, there are some other models that may also raise interests for the research community, *e.g.*, guidance distilled models that uses no classifier-free guidance, and visual autoregreessive models. We present the utility of the synthetic data generated from FLUX-schnell (Labs, 2024) (a distilled diffusion model) and Infinity (Han et al., 2025) (a visual autoregressive model) using CC12M prompts in Figure 12. We include vanilla guidance and prompt expansion for both models.

The Infinity model has the lowest diversity compared to the LDM model series. The LDMv3.5L model has a diversity metric of 4.4 as the lowest, while Infinity can only effectively generate 2 different images on average across prompt lengths. Prompt expansion boosts the diversity metric for all the complexities. Similar to the LDM model series, the diversity of the synthetic images generated using level-1 prompts surpass that of the real data. Interestingly, the decrease of the diversity along the prompt complexity axis is more pronounced than the LDMv1.5 and LDMv3.5L models as shown in Figure 2a. This indicates that the prompt may put more constraints on a visual autoregressive model, *e.g.*, Infinity, when exploring the image space compared to diffusion or flow models. Nevertheless, the Infinity model shows good quality that surpasses the real data and the LDMv1.5 model. It is on par with or a bit lower in terms of aesthetic quality compared to the LDMv3.5L model. Consistency-wise, the Infinity achieves the best compared to all the LDM model series. This may also correlate with the low diversity of the Infinity model as the model is more faithful to the prompt and thus limit the exploration in the image space.

As for the FLUX-schnell model, it performs similarly to the LDMv3.5L model on three utility axes, with a small but noticeable drop in utility, especially on the axes of diversity and consistency. The gap tends to become larger when the prompt complexity increases.

## G  ADDITIONAL RESULTS

### G.1  REFERENCE-BASED METRICS USING IMAGENET-1K CLASS LABELS

Figure 13 shows the reference-based metrics using ImageNet-1k class labels. The findings are similar with the CC12M dataset, as presented in section 4.3.

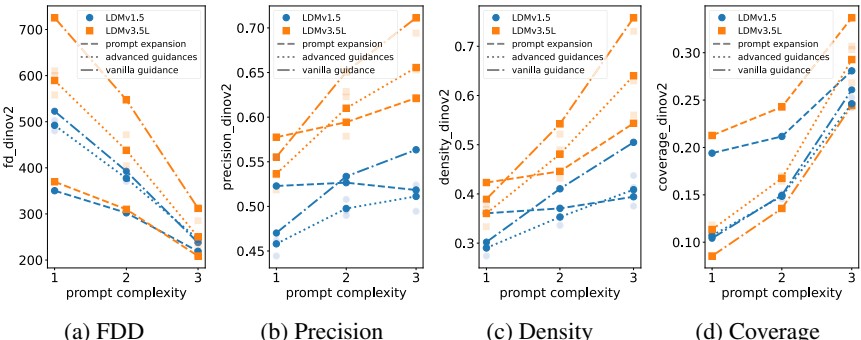

|  |  |  |  |
|:---:|:---:|:---:|:---:|
| (a) FDD | (b) Precision | (c) Density | (d) Coverage |

Figure 13: **Reference-based utility metrics of synthetic data using ImageNet-1k class labels.** FDD, precision, density, and coverage metrics in the DINOv2 feature space for LDMv1.5 and LDMv3.5L with: 1) vanilla guidance (CFG), 2) prompt expansion, and 3) advanced guidance methods, for which transparent markers correspond to different methods and the solid marker is the average over methods. Both advanced guidance methods and prompt expansion lead to better FDD, and improve or match the coverage of vanilla guidance, while sacrificing precision and density in most cases. For LDMv3.5L, we see a more prominent effect of advanced guidance methods and prompt expansion than for LDMv1.5. Advanced guidance methods do not change the trend of the metrics *w.r.t.* prompt complexity, while the prompt expansion shapes the trend of the metrics differently.

## G.2 TRADE-OFF OF CREATIVITY AND FIDELITY WHEN USING INFERENCE-TIME INTERVENTIONS

Figs. 5 and 13 show that using prompt expansion and advanced guidance will result in a drop in precision and density, revealing an under-explored but important trade-off between creativity and fidelity. We suggest cautious usage of T2I models when applying to different downstream applications. We provide a discussion over this trade-off in the following.

**Beneficial Scenarios:**

1. Creative ideation and co-creation. Beyond simple augmentation, prompt expansion acts as an exploratory partner for artists or designers. It uses the LLM's "imagination" to brainstorm novel, high-aesthetic compositions that diverge from the dataset's common tropes.

2. Combating prompt-level mode collapse. For simple prompts (e.g., a dog), T2I models can default to a typical representation (e.g., a golden retriever). Prompt expansion explicitly forces intra-class diversity by generating a set of specific instances, preventing the model from settling on a single, average output.

3. Training for OOD Robustness: "hallucinations" create data that is plausible but statistically rare in the reference set, making it a useful source for training downstream models that are robust to out-of-distribution contexts.

**Detrimental Scenarios:**

1. Bias amplification and stereotype entrenchment: This is a critical risk we will add. The LLM's prior, used for expansion, is not neutral. It can inject societal biases (e.g., related to gender or race for prompts like "a doctor" or "a CEO"), amplifying stereotypes by hardcoding them into the prompt before the T2I model ever runs.

2. Loss of user intent and control. When a user desires generality (e.g., "a simple icon of a bird"), prompt expansion's forced specificity is a failure to respect the user's intent. It overrides their desire for an "average" or "iconic" representation.

### G.3 STATISTICAL SIGNIFICANCE OF THE TRENDS

To ensure the statistical significicnace of the observed trends, we compute the mean and 95% confidence interval of the reference-free metrics across complexities using 5,000 randomly sampled CC12M prompts. The results are shown in Table 8, 9, and 10 for diversity, quality, and consistency respectively. The confidence intervals (95% CI) are narrow, indicating high stability in the measurement. Notably, the intervals between consecutive complexity levels are non-overlapping, confirming that the main trends at each complexity are statistically significant.

Table 8: 95% Confidence Interval for Diversity using CC12M prompts

| complexity | 1 | 2 | 3 | 4 |
|---|---|---|---|---|
| **LDMv1.5** + CFG | 9.228±0.094 | 7.649±0.079 | 6.957±0.067 | 6.483±0.059 |
| + APG | 10.163±0.093 | 8.701±0.081 | 7.921±0.070 | 7.496±0.063 |
| + CADS | 11.112±0.095 | 9.557±0.081 | 8.496±0.072 | 7.902±0.064 |
| + Interval | 11.934±0.089 | 10.378±0.082 | 9.314±0.070 | 8.839±0.066 |
| + Prompt expansion, CFG | 15.074±0.089 | 12.977±0.097 | 10.882±0.092 | 9.725±0.081 |
| **LDMv3.5L** + CFG | 5.597±0.053 | 5.493±0.044 | 4.294±0.032 | 3.966±0.028 |
| + APG | 8.978±0.090 | 7.331±0.079 | 6.089±0.065 | 5.302±0.054 |
| + CADS | 9.864±0.108 | 7.690±0.093 | 5.954±0.067 | 5.214±0.056 |
| + Interval | 10.707±0.088 | 9.463±0.086 | 7.668±0.073 | 6.788±0.064 |
| + Prompt expansion, CFG | 14.671±0.098 | 12.114±0.107 | 9.140±0.100 | 7.672±0.083 |

Table 9: 95% Confidence Interval for Quality using CC12M prompts

| complexity | 1 | 2 | 3 | 4 |
|---|---|---|---|---|
| **LDMv1.5** + CFG | 4.085±0.011 | 4.297±0.013 | 4.335±0.012 | 4.357±0.012 |
| + APG | 4.035±0.011 | 4.213±0.012 | 4.214±0.011 | 4.233±0.011 |
| + CADS | 4.013±0.011 | 4.195±0.012 | 4.239±0.011 | 4.254±0.011 |
| + Interval | 4.017±0.009 | 4.163±0.012 | 4.204±0.011 | 4.213±0.011 |
| + Prompt expansion, CFG | 4.645±0.009 | 4.620±0.009 | 4.472±0.010 | 4.415±0.010 |
| **LDMv3.5L** + CFG | 5.011±0.011 | 5.155±0.012 | 5.016±0.013 | 4.981±0.013 |
| + APG | 5.071±0.009 | 5.179±0.010 | 5.012±0.012 | 4.967±0.012 |
| + CADS | 5.019±0.010 | 5.118±0.011 | 4.977±0.012 | 4.940±0.013 |
| + Interval | 5.042±0.008 | 5.101±0.009 | 4.987±0.009 | 4.924±0.011 |
| + Prompt expansion, CFG | 5.238±0.008 | 5.229±0.009 | 5.110±0.010 | 5.072±0.010 |

Table 10: 95% Confidence Interval for Consistency using CC12M prompts

| complexity | 1 | 2 | 3 | 4 |
|---|---|---|---|---|
| **LDMv1.5** + CFG | 0.9026±0.0026 | 0.8153±0.0029 | 0.7630±0.0025 | 0.7175±0.0022 |
| + APG | 0.8853±0.0028 | 0.8011±0.0030 | 0.7520±0.0026 | 0.7036±0.0023 |
| + CADS | 0.8724±0.0029 | 0.7928±0.0030 | 0.7438±0.0026 | 0.6981±0.0023 |
| + Interval | 0.8573±0.0030 | 0.7788±0.0031 | 0.7313±0.0026 | 0.6822±0.0023 |
| + Prompt expansion, CFG | 0.8072±0.0034 | 0.7130±0.0034 | 0.6902±0.0028 | 0.6502±0.0024 |
| **LDMv3.5L** + CFG | 0.9235±0.0023 | 0.8404±0.0028 | 0.8001±0.0024 | 0.7622±0.0025 |
| + APG | 0.9063±0.0025 | 0.8270±0.0029 | 0.7983±0.0024 | 0.7606±0.0021 |
| + CADS | 0.8546±0.0031 | 0.7881±0.0031 | 0.7636±0.0025 | 0.7251±0.0022 |
| + Interval | 0.8677±0.0029 | 0.7901±0.0031 | 0.7715±0.0025 | 0.7378±0.0022 |
| + Prompt expansion, CFG | 0.8473±0.0031 | 0.7494±0.0032 | 0.7432±0.0026 | 0.7083±0.0023 |

### G.4 CLOSER LOOK AT MODELS AND GUIDANCES

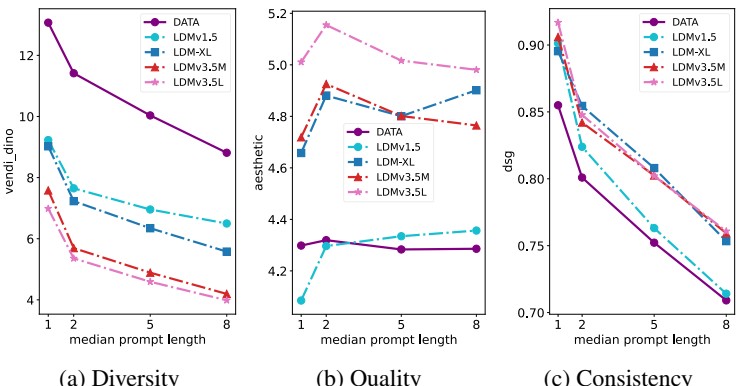

|     |     |     |
| --- | --- | --- |
| (a) Diversity | (b) Quality | (c) Consistency |

Figure 14: **Reference-free utility metrics of synthetic data from different LDM models using CC12M prompts.** We show Vendi score, aesthetic score, and DSG score using different LDM models with CFG. We see a decrease of diversity across model release time – *i.e.*, more recent models tend to have lower diversity. Interestingly, LDMv3.5L model shows less diversity than LDMv3.5M model. When it comes to image quality (aesthetics score), LDMv3.5L exhibits the best performance, followed by LDM-XL and LDMv3.5M. LDMv1.5 shows the worst aesthetic score, which is however similar to that of the dataset. When it comes to image-prompt consistency, all models show relatively high DSG scores. However, LDMv1.5's consistency drops more sharply than the consistency of other models.

**A closer look at different models.** Figure 14 presents reference-free metrics (Vendi, aesthetic, and DSG scores) comparing different LDM models on the CC12M dataset when using CFG guidance. We observe that Vendi score and aesthetics score follow opposite model ranking trends, with LDMv3.5L showing the best aesthetic quality and the lowest diversity performance. As for image-prompt consistency, we observe that LDMv1.5 model falls short from other models by a large margin.

Figure 15 presents reference-based metrics comparing the same models. LDMv3.5M model works surprisingly well in terms of FDD compared to LDMv3.5L, considering that they both show similar reference-free diversity and that LDMv3.5M has a lower aesthetic quality than LDMv3.5L. Looking into precision, density and coverage, we can better understand why this happens. LDMv3.5M model shows similar precision and higher density than LDMv3.5L, while also showing higher coverage, leading to an overall better FDD. Our results on CC12M dataset show that LDMv3.5M may better represent the real data distribution than LDMv3.5L.

**A closer look at advanced guidance methods.** We now shift our focus to study the effect of different advanced guidance approaches on the utility axes of synthetic data. Figure 16 presents the results of reference-free metrics from LDMv3.5L model using CC12M prompts of increasing complexity. All advanced guidance methods lead to substantially higher diversity than the baseline CFG (previously referred to as vanilla guidance), with a stable aesthetic quality across different approaches. This is observed consistently across different prompt complexities. It is worth noting that APG leads to the lowest consistency drops *w.r.t.* CFG. APG primarily addresses the over-saturation problem in synthetic images by reducing the perpendicular direction guidance but still uses the conditional information at all inference steps, which might help the method achieve higher consistency than other the advanced guidance methods. However, CADS and Interval are more aggressive in removing the conditional information during the generation process, leading to overall lower consistencies across prompt complexities. As for reference-based metrics, all advanced guidance methods lead to better FDD. These results suggest that the advanced guidance methods may effectively improve the diversity of synthetic images while maintaining a stable quality and sufficiently good consistency, without sacrificing the overall reference-based metric (FDD). Finally, when contrasting with real data utility, we observe that: 1) None of the advanced guidance techniques bridges the gap between the diversity resulting from vanilla CFG guidance and the real data diversity for LDMv3.5L model. 2) All advanced guidance techniques have similar aesthetics, which is in all cases superior to the

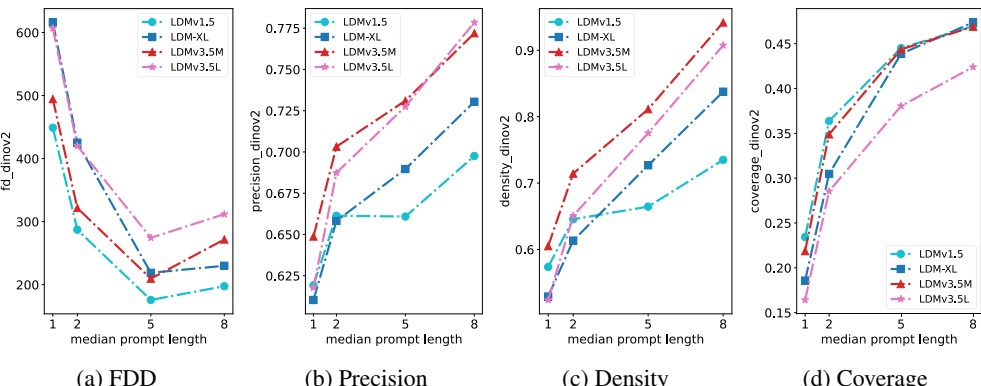

(a) FDD  (b) Precision  (c) Density  (d) Coverage

Figure 15: **Reference-based utility metrics of synthetic data from different LDM models using CC12M prompts.** FDD, precision, density, and coverage metrics in the DINOv2 feature space for different LDM models when using CFG guidance scale 5. For FDD, LDMv1.5 shows the best performance. Interestingly, LDMv3.5M has better FDD than LDMv3.5L. As for precision and density, more recent models tend to have better performances. LDMv3.5M and LDMv1.5 models show similar coverage, with LDM-XL slightly falling behind, and LDMv3.5L presenting the lowest coverage.

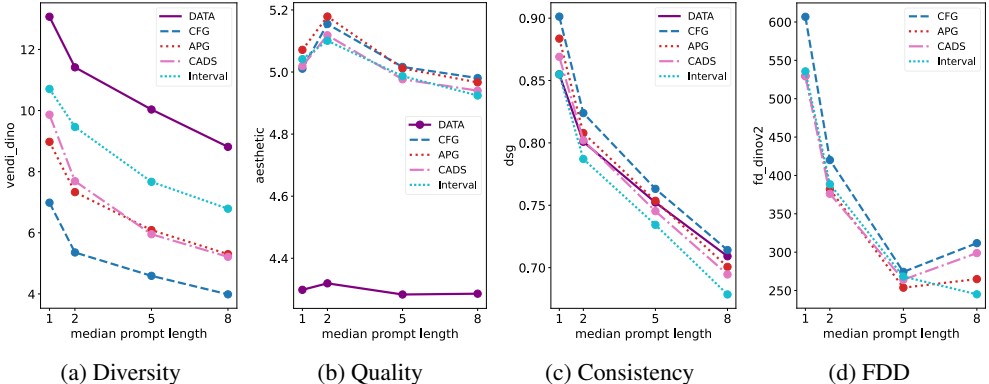

(a) Diversity  (b) Quality  (c) Consistency  (d) FDD

Figure 16: **Effect of guidance methods on the utility of synthetic data from LDMv3.5L using CC12M prompts.** Diversity (Vendi), quality (aesthetics), consistency (DSG), and FD in DINOv2 feature space. All advanced guidance methods lead to higher diversity than the CFG baseline. APG has the smallest negative effect on consistency compared to CADS and Interval guidance. All advanced guidance methods lead to better FDD.

one of the real data. 3) Among advanced guidance methods, APG appears to sacrifice prompt-image consistency the least.

**A closer look at guidance scales.** In this paragraph, we study how guidance scale affects the utility of synthetic data. We present diversity (Vendi score) and aesthetic quality for LDMv1.5 and LDMv3.5L with guidance scales $3.0, 5.0, 7.0$, and $9.0$ in Figure 17. Perhaps unsurprisingly, we observe that increasing the guidance scale results in decreasing the diversity of synthetic images. However, the diversity captured by Vendi score increases for LDMv3.5L model when using very large guidance scale (*e.g.*, 9.0 in Figure 17b). We hypothesize that this increase in Vendi score is due to the oversaturation in the synthetic images and the color range exceeding the color jittering range employed in the data augmentation process of DINOv2, making the DINOv2 features color-sensitive. For different models and different metrics, the sensitivity to the guidance scale varies. For example, LDMv1.5 is more sensitive towards diversity while LDMv3.5L is more sensitive towards aesthetic quality as the guidance scale changes. Interestingly, LDMv1.5 has better aesthetic quality with higher guidance scales while LDMv3.5L shows the opposite trend.

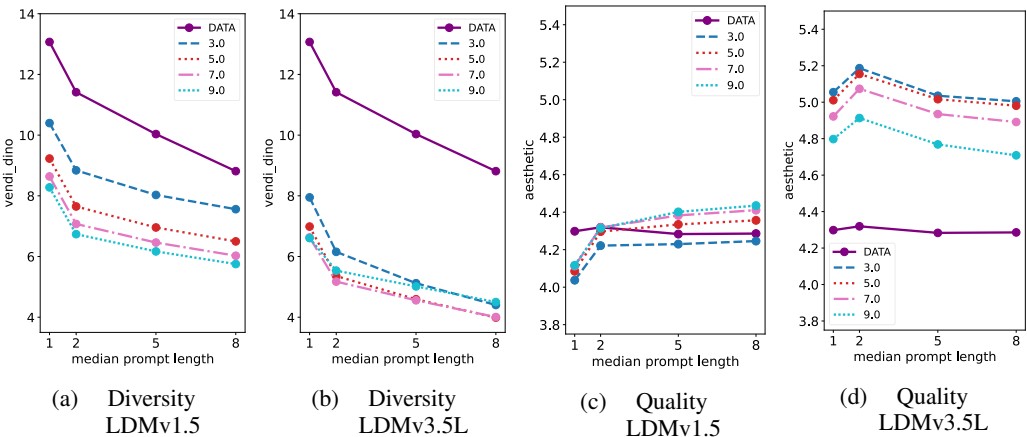

Figure 17: **Effect of guidance scale on diversity (Vendi) and quality (aesthetic) of synthetic images from CC12M prompts.** Metrics are computed in the DINOv2 feature space. For LDMv1.5, the diversity decreases as we increase the guidance scale, while for LDMv3.5L, the diversity first decreases and then increases again when using the highest guidance scale. LDMv1.5 shows better aesthetic quality when increasing the guidance scale while LDMv3.5L shows the opposite trend.

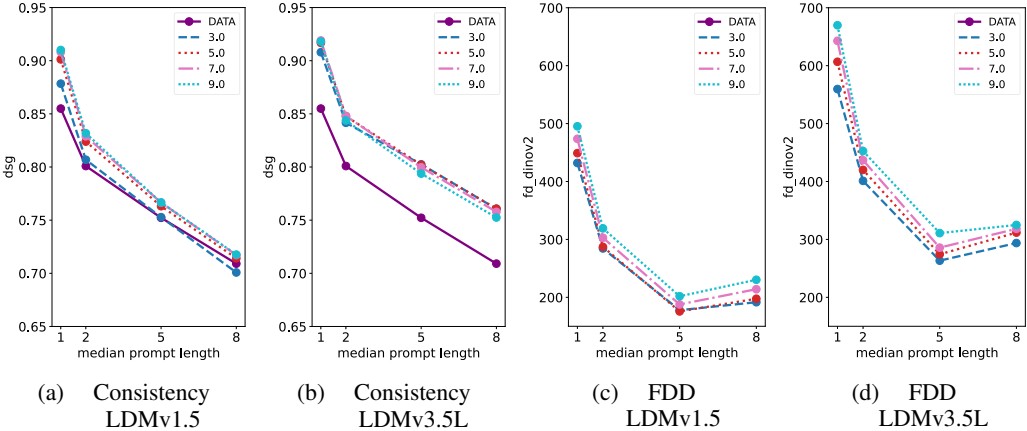

Figure 18: **Effect of guidance scale on image-prompt consistency (DSG) and FDD of synthetic images from CC12M prompts.** DSG score is not sensitive to guidance scale, especially for LDMv3.5L model. For LDMv1.5 model, the consistency slightly increases when employing larger guidance scales. As for FDD, both LDMv1.5 and LDMv3.5L models exhibit increase of FDD when generating images with larger guidance scale.

We also depict consistency and FDD metrics using different guidance scales in Figure 18. Image-prompt consistency does not appear too sensitive to guidance scale, especially for LDMv3.5L. For LDMv1.5, the consistency slightly increases when employing larger guidance scales. As for FDD, both LDMv1.5 and LDMv3.5L exhibit increases of FDD when generating images with larger guidance scales.

## H  ADDITIONAL QUALITATIVE SAMPLES

In Figure 19, we show qualitative samples for LDMv3.5L model using vanilla guidance (CFG), CADS, and prompt expansion with prompts extracted from CC12M. Notably, we observe that as we increase prompt complexity, the generations appear closer to the real data and exhibit lower diversity, slightly lower prompt consistency, and similar image aesthetics. Moreover, inference-time

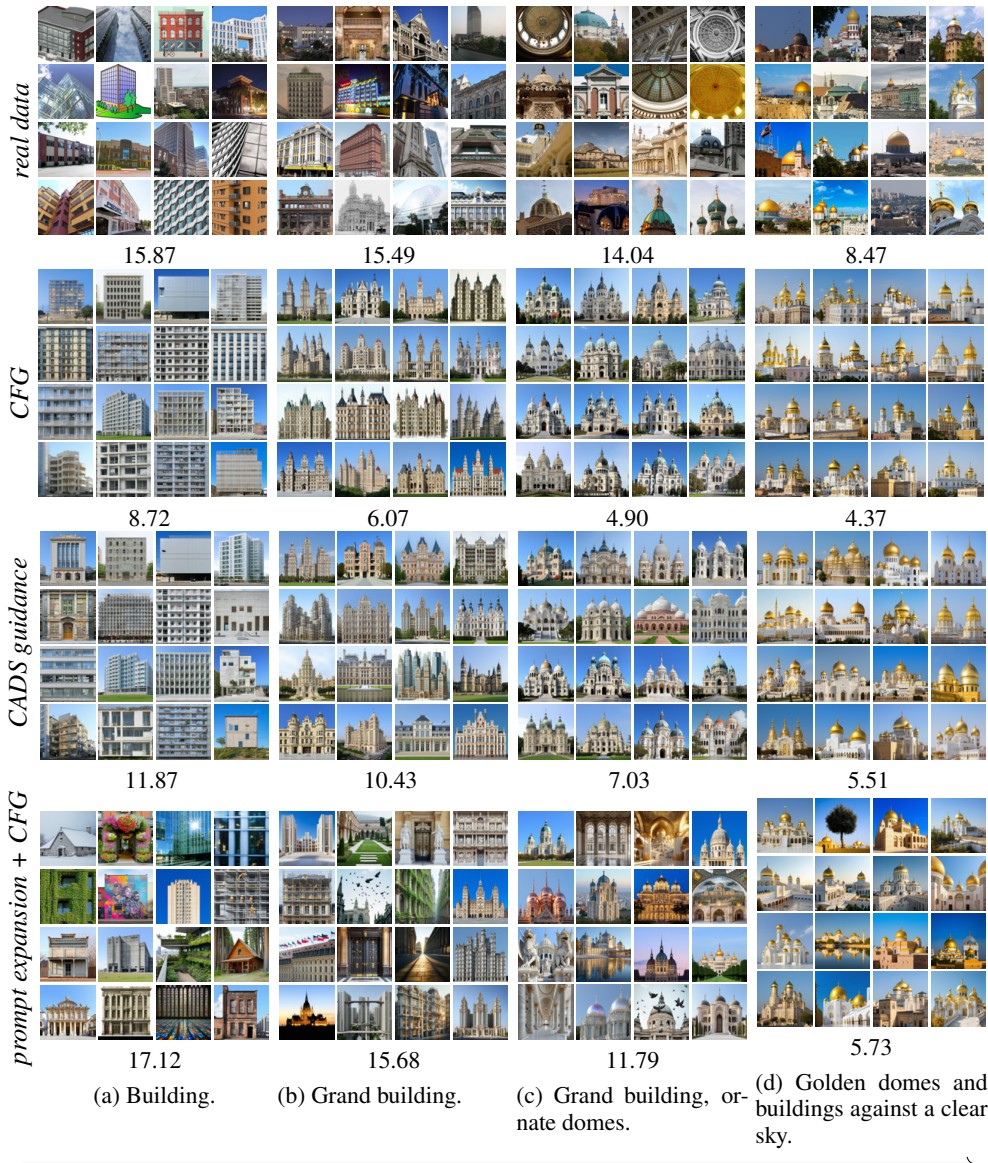

*prompt complexity increasing*

Figure 19: **Synthetic images across prompt complexity (row-wise) and sampling methods (column-wise).** The number below each mosaic corresponds to the diversity metric (Vendi score) of that set of images. Diversity decreases both visually and quantitatively as we increase prompt complexity. Advanced sampling methods (CADS and prompt expansion) improve the diversity of synthetic data. Image quality is not significantly affected by the prompt complexity, but prompt consistency decreases as we increase prompt complexity. CFG: classifier-free guidance.

interventions and their combinations (rows 3, 4) result in synthetic data with higher diversity than the CFG baseline (row 2).

We show qualitative samples in Figure 20 for LDMv3.5L model using vanilla guidance (CFG), CADS, and prompt expansion with class labels extracted from ImageNet-1k. According to WordNet (Fellbaum, 1998), ``stringed instrument'' includes harp, violin, cello, guitar, piano, etc., ``guitar'' includes acoustic guitar and electric guitar.

We also show more qualitative samples in Figures 21 to 28 using all sampling intervention methods, generated by all four T2I models (LDMv1.5, LDM-XL, LDMv3.5M, and LDMv3.5L), complement-

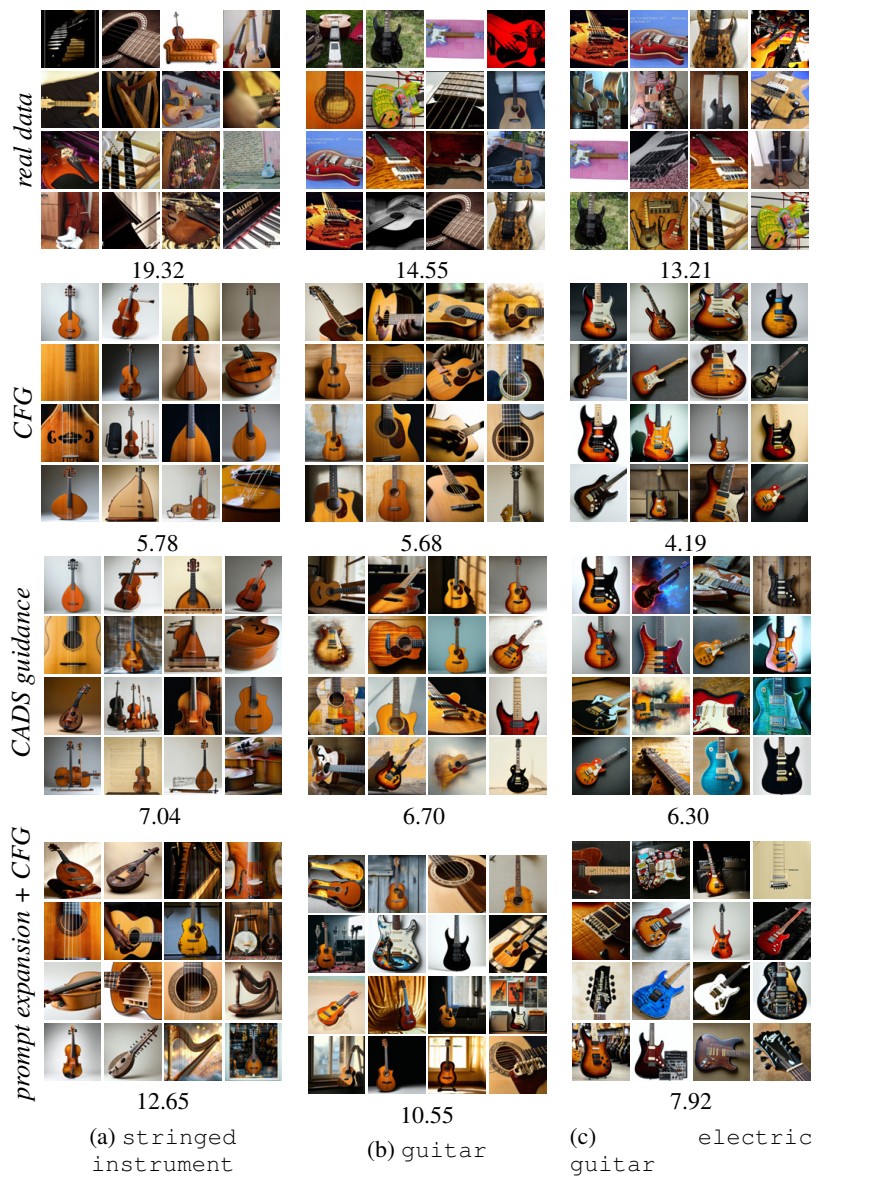

Figure 20: **Synthetic images across prompt complexity (row-wise) and sampling methods (column-wise) using prompts from ImageNet-1k.** The number below each mosaic corresponds to the diversity metric (Vendi score) of that set of images. Diversity decreases both visually and quantitatively as we increase prompt complexity. Advanced sampling methods (CADS and prompt expansion) improve the diversity of synthetic data. Image quality is not significantly affected by the prompt complexity, but prompt consistency decreases as we increase prompt complexity. CFG: classifier-free guidance.

ing the results in Figure 19. We observe qualitatively that prompt expansion and advanced guidance methods lead to more diverse images than the vanilla guidance.

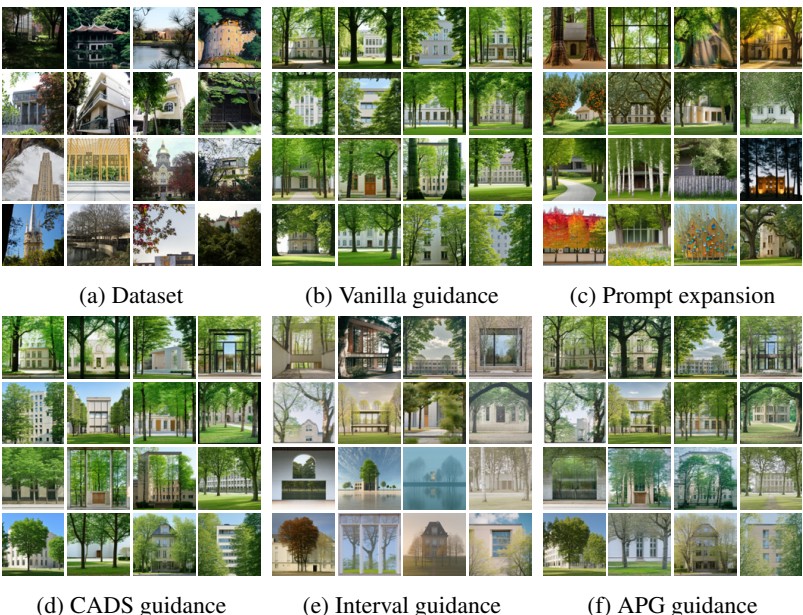

Figure 21: **Qualitative visuals for the prompt ``Trees frame a peaceful building''
using the LDMv3.5L model with different sampling settings.**

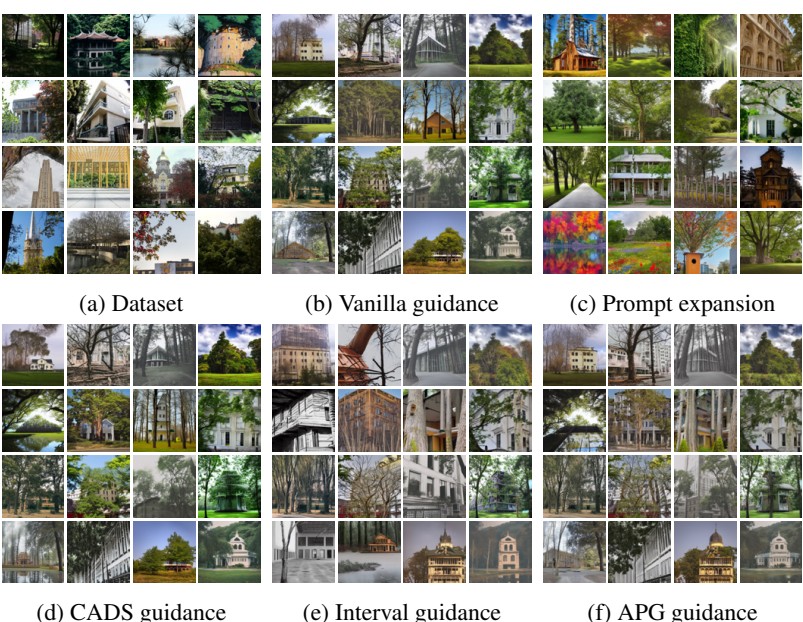

Figure 22: **Qualitative visuals for the prompt ``Trees frame a peaceful building''
using the LDMv1.5 model with different sampling settings.**

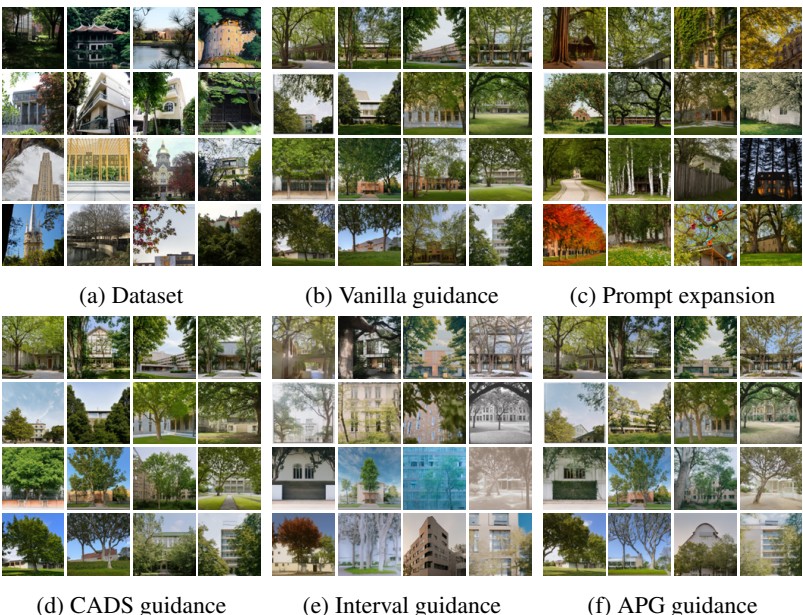

Figure 23: **Qualitative visuals for the prompt ``Trees frame a peaceful building''
using the LDMv3.5M model with different sampling settings.**

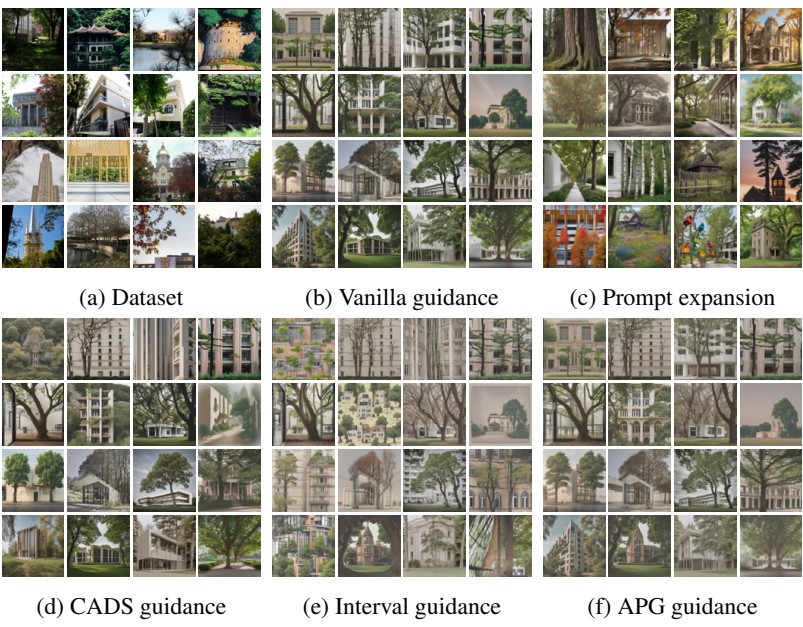

Figure 24: **Qualitative visuals for the prompt ``Trees frame a peaceful building''
using the LDM-XL model with different sampling settings.**

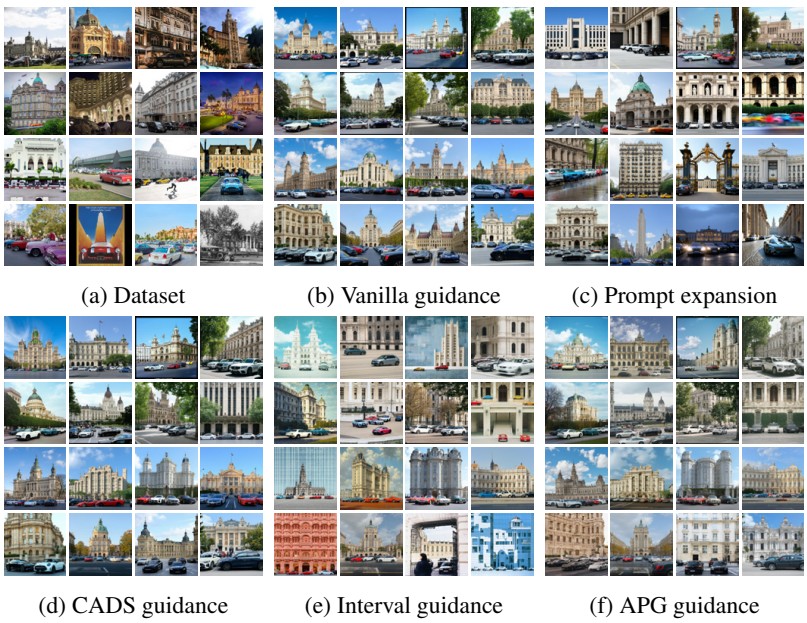

Figure 25: **Qualitative visuals for the prompt ``Cars and a grand building'' using the LDMv3.5L model with different sampling settings.**

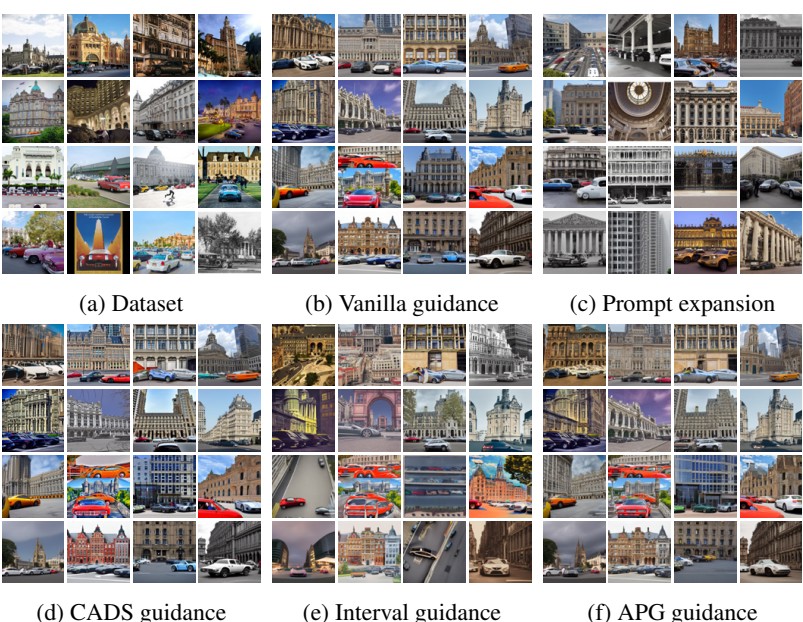

Figure 26: **Qualitative visuals for the prompt ``Cars and a grand building'' using the LDMv1.5 model with different sampling settings.**

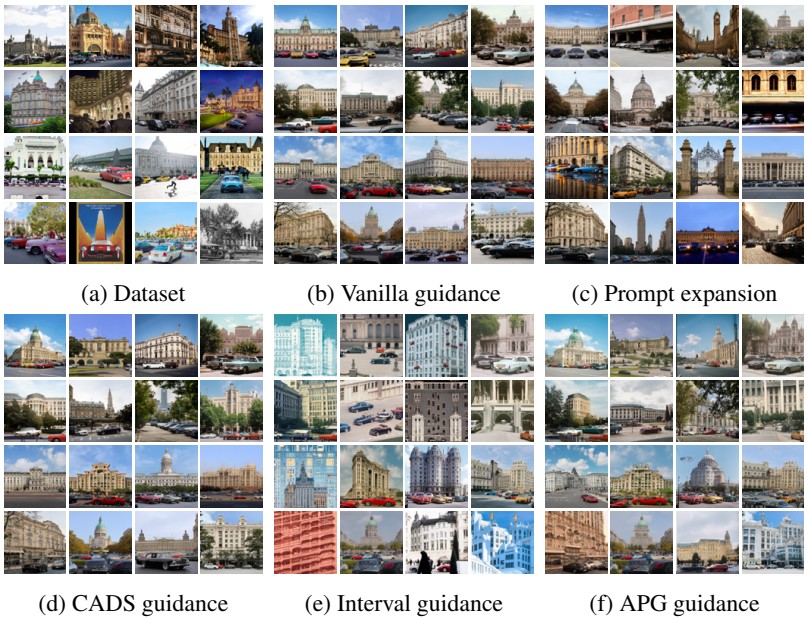

Figure 27: **Qualitative visuals for the prompt ``Cars and a grand building'' using the LDMv3.5M model with different sampling settings.**

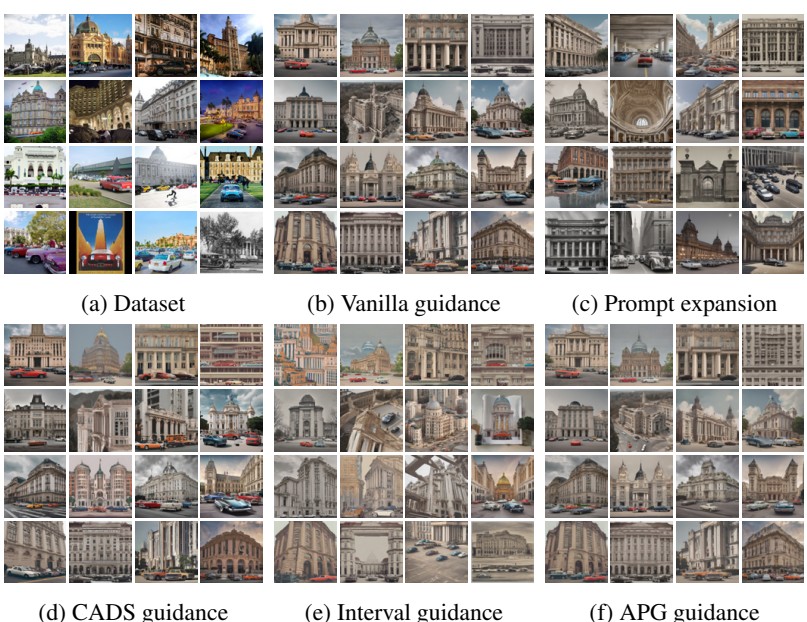

Figure 28: **Qualitative visuals for the prompt ``Cars and a grand building'' using the LDM-XL model with different sampling settings.**

