# OpenReview forum: "The Intricate Dance of Prompt Complexity, Quality, Diversity and Consistency in T2I Models"
_ICLR.cc/2026/Conference — ICLR 2026 Poster_

### Official Review · Reviewer_CGJq · 2025-10-29

**Soundness:** 2
**Presentation:** 2
**Contribution:** 2
**Rating:** 6
**Confidence:** 4

**Summary:**

This paper analyzes how prompt complexity affects the quality, diversity, and text-image alignment of text-to-image (T2I) generations. Through synthetic and large-scale experiments, the authors show that generalizing to simpler (more general) prompts is inherently harder for diffusion models. Increasing prompt complexity reduces diversity and consistency but narrows the gap to real data. Among inference-time methods, prompt expansion using large language models to enrich prompts consistently improves image quality and diversity, and combining it with advanced guidance yields the best trade-offs.

**Strengths:**

- It is interesting to investigate the relationship between prompt complexity and the utility of generative models, including quality, diversity, and text-image alignment.

- The paper provides several insightful observations, such as the generalization asymmetry between simple and complex prompts and the non-monotonic improvement of quality as complexity increases.

- The authors propose an improved sampling strategy that combines prompt expansion with advanced diversity samplers (e.g., APG), based on the observation that moderately complex prompts can enhance generation quality.

**Weaknesses:**

- **Lack of new messages.** Apart from the findings on generalization asymmetry and non-monotonic quality trends, most conclusions (e.g., that LLM-based prompt expansion increases diversity) are fairly predictable or already known.

- **Weak application value.** Except for the combined sampler (e.g., prompt expansion + APG), the study remains largely descriptive and provides limited guidance for improving model performance.

- **Clarity and focus.** The paper covers a large number of datasets, models, and metrics, which sometimes dilutes the main narrative and makes it harder to pinpoint the central contribution.

- **Limited theoretical depth.** Although the paper provides a toy Gaussian experiment and a theoretical discussion, the analysis remains relatively shallow and does not fully explain why diffusion models exhibit the observed asymmetry beyond intuitive reasoning.

- **Evaluation bias toward LDM family.** The study mainly focuses on LDM variants, leaving uncertainty about whether the conclusions generalize to non-diffusion architectures or transformer-based T2I models.

- **No user evaluation.** The work focuses entirely on metric-based analyses without human validation to confirm the practical implications of prompt complexity.

**Questions:**

See weaknesses.

---

> ### Author Response · Authors · 2025-11-22
> **Response to Reviewer CGJq - Part 1**
>
> Thank you for detailed review and for acknowledging the interestingness of our investigation, the insightful observations, and the improved sampling strategy for better synthetic data utility.
>
> + **Lack of new messages**
>
> Thank you for acknowledging the novelty of findings on generalization asymmetry and non-monotonic quality trends.
>
> First, we would like to argue the importance of **transitioning from intuition to evidence over comprehensive datasets, models and inference-time interventions.** "Intuition" is often unquantified, contradictory, and inaccessible to newcomers. Our work is *the first to systematically and quantitatively* evaluate these trade-offs along *with prompt complexities*. We believe that a comprehensive evaluation is essential to build scientific knowledge, confirming or refuting intuitions that are otherwise built on limited or anecdotal evidence.
>
> Second, we are actually surprised by many of the findings revealed in our work and we highlight some of them as follows:
>
> (1) As you noted, the trend of utilities are non-linear, and especially the aesthetic score exhibits a slope steeper towards shorter prompt lengths and more gradual for longer ones (Fig 3 c), showing *an asymmetry of prompt length generalization*.
>
> (2) Contrary to our intuition that very long prompts should collapse diversity, we find that *diversity plateaus as prompt length increases*, as shown in DCI. This suggests an *inherent “lower bound of diversity”* in sampling.
>
> (3) We find that *optimizing for reference-free metrics harms distributional fidelity*. As shown in our Frechet Distance measurements, prompt expansion and *newer models (e.g., LDMv3.5L)* that achieve higher reference-free metrics actually *degrade in distributional alignment* with real data. This implies that “chasing” these scores drives the generative distribution away from the real data manifold, suggesting *a cautious usage of synthetic data generated from T2I models for downstream applications*. We also discuss this point in the response to reviewer PuV8.
>
> To our knowledge, the systematic, quantitative evidence on utility of synthetic data generated by T2I models over prompt complexity perspective is novel and has not been reported. We are happy to incorporate related works that have reported similar results into our manuscript if you have specific suggestions.
>
> + **Weak application value**
>
> Thank you for pointing out this. We humbly argue that proposing a benchmarking method itself is a primary contribution and has its strong application value, *offering the community a standardized tool for assessment where none previously existed*. The comprehensive, rigorous, and reproducible evaluation framework is a contribution acknowledged by all three other reviewers QZqM, N9uF and PuV8.
>
> Further, with the results derived from our evaluation pipeline, we summarize the following suggestions for future model training and inference:
>
> 1. **Reducing conditional ambiguity (Training Insight):** Incorporating LLM-derived semantics during training is a logical next step supported by our analysis. Short, general prompts create a high-entropy conditional distribution (mapping one text to a vast space of possible images), which is difficult for the model to learn. *Fine-grained, LLM-augmented detailed descriptions reduce this ambiguity, sharpening the conditional distribution and simplifying the learning objective.*
>
> 2. **Bridging the gap via inference:** Ideally, inference prompts should match the distribution of training prompts. However, because training data for large-scale models is often *unavailable or proprietary*, and retraining is computationally expensive, “training-time” solutions are not always accessible to practitioners. Our work demonstrates that *combining inference-time interventions* can approximate this training-inference alignment, improving the generation diversity and quality from T2I models.
>
> + **Clarity and focus**
>
> Thank you for acknowledging the comprehensiveness of our paper, with a *wide coverage of datasets, models, and metrics*. We were concerned that if we used only one model or dataset, our findings would be dismissed as an artifact. Therefore, the comprehensive study across many models with different architecture and backbones (LDMv1.5, LDM-XL, LDMv3.5, Flux, and Infinity), datasets (CC12M, ImageNet, DCI), and metrics was necessary to demonstrate the generality of our findings. We believe this breadth shows that the utility trade-offs as a function of prompt complexity are a fundamental property of current T2I models, not a quirk of a single setup. **With the clarifications in previous bullet points, we hope that we clarify the novelty and contribution of our paper, with a clear focus.** We will also revise the introduction and provide a revised version that incorporates all the points raised during the rebuttal.

---

> ### Author Response · Authors · 2025-11-22
> **Response to Reviewer CGJq - Part 2**
>
> + **Limited theoretical depth**
>
> Thank you for raising this point. We use the synthetic experiment as a **motivation** to emphasize the importance of studying the axis of **prompt complexity** in the diffusion / flow model setting. The synthetic experiments clearly show **an asymmetry of difficulty** w.r.t. conditioning complexity. We do not intend to directly apply these theoretical predictions to real T2I models, but rather use them as an intuitive “clean room” that provides a hint when similar observations are witnessed in real settings.
>
> We further discuss during the rebuttal **the theoretical-empirical gaps** because of the independence assumption we made for simplification as stated by reviewer QZqM. We will incorporate the detailed discussion with reviewer QZqM into the revised manuscript.
>
> + **Evaluation bias toward LDM family**
>
> Thank you for raising this concern. We want to clarify that our study does **cover a wider range of models** than just the UNet-based LDM family. We present in lines 252 - 265 all the models (Stable Diffusion 1.5, XL, and 3.5 Large and Medium, along with Flux-schnell and Infinity) evaluated in our paper. Specifically, LDMv3.5 Large and Medium, also Flux-schnell are **flow models** using **MM-DiT (transformer-based)** architecture. Further, because of the page limit, we report the evaluation results of Flux-schnell (a flow model using transformer-based architecture, distilled for faster guidance inference) and **Infinity (a visual autoregressive model which does not use diffusion process)** in Appendix F. Thus, our analysis covers a wide range of models including non-diffusion processes and transformer-based T2I models and we report results comprehensively.
>
> + **No user evaluation**
>
> Thank you for acknowledging the importance of human evaluation in our work, which is also appreciated by reviewer QZqM. We would like to draw your attention to **line 313 where we mentioned that our human evaluation over diversity metric is presented in Appendix E.** This human evaluation shows that Vendi Score in Dinov2 feature space strongly correlates with human-perceived diversity, **validating the automatic diversity metric.**
>
> During the rebuttal, we also conducted human evaluation over the **consistency metric** following the suggestions of you and reviewer PuV8. Here are the human evaluation results.
>
>
> **Stage 1: Decomposition**
>
> We give the guideline on how the prompt is decomposed into atomic questions along with an example of decomposition presented to human annotators. Given a prompt, we present the prompt along with the decomposed questions generated by LLM to human evaluators, and ask them to check whether each question is a valid question according to decomposition rules. We ask 4 human annotators, and each of them annotates 25 prompts, yielding 100 sets of answers in total. Among all the decomposed questions, the percentage of correctly decomposed questions is **95.54%, which effectively demonstrates the validity of the decomposed questions using LLM automatic generation.**
>
> **Stage 2: VQA model reliability**
>
> Given the generated images from the original long prompt, we ask the human annotators to answer each decomposed question generated by LLM from the original long prompt. We ask 10 annotators and each evaluates 50 generated images using DCI prompts and gather in total 500 evaluation results. The human evaluation is small compared to the large-scale automatic evaluation, but it shows the same decreasing trend and **a strong correlation (Pearson correlation 0.89 and R2 0.68) between human evaluation results and automatic VLM results**. Table 1 shows the human evaluation results for Stage 2.
>
> Table 1 DSG metrics computed using human evaluation and automatic VLM
> | \# words   | 10                         | 13   | 18   | 23   | 28   | 35   | 41   | 50   |
> | ---------- | -------------------------- | ---- | ---- | ---- | ---- | ---- | ---- | ---- |
> | \# samples | 58                         | 54   | 66   | 61   | 65   | 66   | 65   | 75   |
> | Human DSG  | 0.92                       | 0.90 | 0.85 | 0.87 | 0.83 | 0.82 | 0.81 | 0.81 |
> | VQA DSG    | 0.95                       | 0.89 | 0.81 | 0.86 | 0.85 | 0.81 | 0.79 | 0.82 |
>
> Thank you again for your insightful review. We are happy to discuss further if you have any follow up question.

---

> ### Comment · Reviewer_CGJq · 2025-11-26
>
> Thank you for the rebuttal. The concerns have been addressed, and I will maintain my current acceptance stance.

---

> > ### Author Response · Authors · 2025-11-30
> >
> > Thank you very much for your reply. We are glad to know that our rebuttal successfully addressed your concerns! Thank you again for your helpful suggestions to improve our paper.

---

### Official Review · Reviewer_PuV8 · 2025-10-31

**Soundness:** 4
**Presentation:** 4
**Contribution:** 3
**Rating:** 8
**Confidence:** 3

**Summary:**

This paper presents a systematic analysis of how prompt complexity influences the quality, diversity, and consistency of synthetic data from text-to-image models. It establishes that increasing prompt complexity reduces diversity and consistency while improving the realism of generated images. The authors provide theoretical insight, showing that models struggle to generalize to general prompts, and introduce a comprehensive benchmarking framework for evaluation. Their large-scale empirical study identifies prompt expansion combined with advanced guidance like APG as the most effective approach for achieving optimal utility trade-offs in synthetic data generation.

**Strengths:**

1.Introduces the novel problem of systematically evaluating "prompt complexity" in T2I models, combining theoretical insight ("OR" vs. "AND" generalization) with practical interventions.

2.Methodologically rigorous, featuring a well-designed benchmarking framework, compelling synthetic experiments, and a large-scale, comprehensive evaluation across models and datasets.

3.Exceptionally well-structured and clearly written, with a logical narrative flow and effective figures that make the complex study accessible and reproducible.

4.Provides crucial guidance for synthetic data generation and model evaluation, revealing fundamental trade-offs and setting an important agenda for future T2I model development.

**Weaknesses:**

1.The paper relies on the DSG score for consistency evaluation but does not critically address its known limitations with very long, complex prompts (as in the DCI dataset). As prompt length increases, the VQA models underlying DSG may themselves fail, potentially conflating model inconsistency with VQA model error. A targeted analysis, such as a human evaluation on a subset of long prompts to verify DSG's correlation with human judgment in this regime, is needed to ensure the reported consistency drop is reliable.

2.While prompt expansion is highlighted as a powerful intervention, the paper provides a limited discussion of its significant downside: it actively moves generations outside the support of the reference real data (as shown by reduced precision/density). This is a critical trade-off that is under-explored. The work would be improved by a deeper analysis of when this "hallucination" is beneficial (e.g., for data augmentation) versus detrimental (e.g., for faithful dataset replication), framing it not just as a boost in diversity but as a fundamental shift in the data distribution.

**Questions:**

1.The authors rely on the DSG score for consistency evaluation. Could they clarify how they ensured this metric's validity for the very long, complex prompts from the DCI dataset? Providing a small-scale human evaluation to correlate with the DSG scores on these long prompts would help solidify the consistency findings.

2.The authors note a drop in consistency for highly complex prompts. Could they comment on the semantic plausibility of these generations? Is the failure primarily in missing attributes, or also in generating globally incoherent scenes?

---

> ### Author Response · Authors · 2025-11-22
> **Response to Reviewer PuV8 - Part 1**
>
> Thank you very much for your constructive review. We are really grateful for your acknowledgement on the novelty of our research problem, the rigorous methodology of evaluation framework, the exceptionally well-structured and clear writing, and the crucial guidance that our work provides for future research.
>
> + **Weakness and questions about DSG score**
>
> Thank you for raising your concern about the DSG score as for consistency evaluation. Our choice of the DSG score (also discussed in lines 457 - 462) was based on its specific mechanism, which is designed to mitigate this exact issue. The VQA model in DSG **does not directly answer the full, complex prompt**. Instead, DSG uses a two-stage process:
>
> 1. **Decomposition:** An LLM (Gemma-3-27B-Instruct, the largest model in the series released in 2025) decomposes the single complex prompt (e.g., "A white dog is playing on the grassy field.") into a set of *simple, atomic questions* (e.g., "Is there a dog?", "Is the dog white?", "Is the field grassy?", …).
>
> 2. **VQA:** The VQA model (default as in DSG score paper) is then only required to *answer these simple, unit-level yes/no questions*, a task it can perform more reliably compared to answering a single long complex question.
>
> To further validate our automatic evaluated results with DSG score, we conduct a **human evaluation** for both stages.
>
> **Stage 1: Decomposition**
>
> We give the guideline on how the prompt is decomposed into atomic questions along with an example of decomposition presented to human annotators. Given a prompt, we present the prompt along with the decomposed questions generated by LLM to human evaluators, and ask them to check whether each question is a valid question according to decomposition rules. We ask 4 human annotators, and each of them annotates 25 prompts, yielding 100 sets of answers in total. Among all the decomposed questions, the percentage of correctly decomposed questions is **95.54%, which effectively demonstrates the validity of the decomposed questions using LLM automatic generation.**
>
> **Stage 2: VQA model reliability**
>
> Given the generated images from the original long prompt, we ask the human annotators to answer each decomposed question generated by LLM from the original long prompt. We ask 10 annotators and each evaluates 50 generated images using DCI prompts and gather in total 500 evaluation results. The human evaluation is small compared to the large-scale automatic evaluation, but it shows the same decreasing trend and **a strong correlation (Pearson correlation 0.89 and R2 0.68) between human evaluation results and automatic VLM results**. Table 1 shows the human evaluation results for Stage 2.
>
> Table 1 DSG metrics computed using human evaluation and automatic VLM
> | \# words   | 10                         | 13   | 18   | 23   | 28   | 35   | 41   | 50   |
> | ---------- | -------------------------- | ---- | ---- | ---- | ---- | ---- | ---- | ---- |
> | \# samples | 58                         | 54   | 66   | 61   | 65   | 66   | 65   | 75   |
> | Human DSG  | 0.92                       | 0.90 | 0.85 | 0.87 | 0.83 | 0.82 | 0.81 | 0.81 |
> | VQA DSG    | 0.95                       | 0.89 | 0.81 | 0.86 | 0.85 | 0.81 | 0.79 | 0.82 |
>
> **On the question about the semantic plausibility of the generations**
>
> The failure is primarily in **missing attributes**, but not in generating globally incoherent scenes. Our prompt consistency analysis, supported by the decomposed outputs of the DSG score, indicates that the failure is mostly in missing attributes or objects. Further, the aesthetic quality, which is based on human preference data, remains in general stable but a slight decrease when extending to highly complex prompts. This indicates that the scenes generated are in general still globally coherent and aligned with human perceptual preferences.

---

> ### Author Response · Authors · 2025-11-22
> **Response to Reviewer - Part 2**
>
> + **Deeper discussion about distributional shift by prompt expansion**
>
> Thank you for this excellent and constructive point. We agree that this trade-off is an interesting finding of our work and its implications deserve a deeper analysis. We also identified and discussed this phenomenon. In lines 398 - 406, we noted: “...The improvements in coverage come... at the expense of precision and density. This is perhaps expected as... prompt expansion may include details... that are not present in the real images, therefore deviating the generation process from the reference data…”
>
> Our current discussion correctly identifies the mechanism for this drop in precision. In the following, we expand this section to provide **a much deeper, more comprehensive discussion** of this "fidelity vs. exploration" trade-off.
>
> **Beneficial Scenarios:**
>
> 1. **Creative ideation and co-creation.** Beyond simple augmentation, prompt expansion acts as an exploratory partner for artists or designers. It uses the LLM's “imagination” to brainstorm novel, high-aesthetic compositions that diverge from the dataset's common tropes.
>
> 2. **Combating prompt-level mode collapse.** For simple prompts (e.g., a dog), T2I models can default to a typical representation (e.g., a golden retriever). Prompt expansion explicitly forces intra-class diversity by generating a set of specific instances, preventing the model from settling on a single, average output.
>
> 3. **Training for OOD Robustness:** “hallucinations” create data that is plausible but statistically rare in the reference set, making it a useful source for training downstream models that are robust to out-of-distribution contexts.
>
> **Detrimental Scenarios:**
>
> 1. **Bias amplification and stereotype entrenchment:** The LLM's prior, used for expansion, is not neutral. It can inject societal biases (e.g., related to gender or race for prompts like "a doctor" or "a CEO"), amplifying stereotypes by hard-coding them into the prompt before the T2I model ever runs.
>
> 2. **Loss of user intent and control.** When a user desires generality (e.g., “a simple icon of a bird”), prompt expansion’s forced specificity is a failure to respect the user's intent. It overrides their desire for an “iconic” representation.
>
> This expanded discussion will be added to lines 398-406 and will significantly strengthen the paper's contribution by providing a much-needed critical perspective on when and why to use this powerful intervention. We are very grateful to you for this suggestion, which has pushed us to explore this important nuance more deeply.
>
> Thank you again for your supportive and constructive review. We are happy to follow up the discussion if you have further questions.

---

### Official Review · Reviewer_N9uF · 2025-11-01

**Soundness:** 3
**Presentation:** 3
**Contribution:** 2
**Rating:** 2
**Confidence:** 4

**Summary:**

The paper studies how prompt complexity (length/specificity) affects the quality, diversity, and prompt-consistency of images generated by text-to-image (T2I) models. It first uses a toy Gaussian setup with derivations to argue that generalizing from fine-grained training prompts to more general prompts is intrinsically harder (likelihood weighting is missing in standard diffusion), then introduces a benchmarking framework that builds multi-complexity prompts from CC12M, ImageNet-1k, and DCI and evaluates several open T2I models and inference-time interventions. Large-scale experiments show: increasing complexity reduces diversity and consistency, while aesthetics first rise then fall as prompts lengthen; prompt expansion improves diversity/aesthetics but may move generations off the real-data manifold; combining expansion with advanced guidance yields the best trade-offs.

**Strengths:**

1. Writing & clarity. The paper is clearly written and easy to follow, with a logical flow (synthetic intuition → framework → large-scale evaluation), helpful figures, and explicit definitions of each utility axis and metric. Reproducibility details (datasets, hyperparameters, code note) are thorough.

2. Comprehensive, reproducible evaluation framework. The authors propose a general framework (captioning → pairing → cross-complexity alignment → sampling → generation) that enables fair comparison between real and synthetic data across complexities, and they test multiple model families (LDM v1.5/v3.5, SDXL, etc.) and guidance strategies (CFG, CADS, Interval, APG) plus prompt expansion, with both reference-free (Vendi, aesthetics, DSG) and reference-based (FDD, precision/density/coverage) metrics—this breadth is valuable for practitioners.

**Weaknesses:**

1. Findings mostly expected; limited novelty. Many conclusions—e.g., higher prompt complexity reduces diversity/consistency, prompt expansion increases diversity but can harm faithfulness—align with common practitioner intuitions and prior observations.

2. No concrete method to handle complexity. The work is primarily evaluative. It surfaces that general prompts are hard because diffusion lacks likelihood weighting and that expansion/guidance can trade off axes—but it stops short of proposing a principled, train-time solution.

**Questions:**

Your work studies semantic composition in prompts. A general question is, could LLM-derived semantics be used during training to improve robustness to complex or general prompts?

Also please see the weaknesses part.

---

> ### Author Response · Authors · 2025-11-22
> **Response to Reviewer N9uF - Part 1**
>
> Thank you very much for your insightful review. We are grateful for your acknowledgement on our **writing clarity**, and **comprehensive, reproducible evaluation framework**, covering a wide range of models, inference-time methods, datasets, and metrics. We are glad that you found **the breadth of our study valuable for practitioners.**
>
> We would like to answer to the reviewer’s two main weaknesses and their question, as we believe our paper’s core contributions may have been mis-stated.
>
> + **Findings mostly expected; limited novelty**
>
> We respectfully disagree that the findings are "mostly expected" or that the work lacks novelty. We posit that the novelty and scientific contribution of our work are three-fold:
>
> 1. **A novel evaluation framework.** As the reviewer and many other reviewers (QZqM and PuV8) kindly noted as a strength, we propose a *rigorous, systematic, and reproducible evaluation framework* that compares the utility of generated images across prompt complexities for open-ended generative models, also against real images. This methodology is *a primary contribution in itself*, offering the community a standardized tool for assessment where none previously existed.
>
> 2. **Transitioning from intuition to evidence over comprehensive datasets, models and inference-time interventions.** We argue that “Practitioner intuition” is often unquantified, contradictory, and inaccessible to newcomers. Our work is *the first to systematically and quantitatively* evaluate these trade-offs along *with prompt complexities*. We believe that a comprehensive evaluation is essential to build scientific knowledge, confirming or refuting intuitions that are otherwise built on limited or anecdotal evidence.
>
> 3. **Non-trivial and unexpected findings.** We are actually surprised by many of the findings revealed in our work.
>
> (1) The trend of utilities are non-linear, and especially the aesthetic score exhibits a slope steeper towards shorter prompt lengths and more gradual for longer ones (Fig 3 c), showing *an asymmetry of prompt length generalization*.
>
> (2) Contrary to our intuition that very long prompts should collapse diversity, we find that *diversity plateaus as prompt length increases*, as shown in DCI. This suggests an *inherent “lower bound of diversity”* in sampling.
>
> (3) We find that *optimizing for reference-free metrics harms distributional fidelity*. As shown in Figs. 5 and 13, prompt expansion degrades *precision and density* while newer models (e.g., LDMv3.5L) degrade in *frechet distance* with real data. This implies that “chasing” these scores drives the generative distribution away from the real data manifold, suggesting *a cautious usage of synthetic data generated from T2I models for downstream applications*. We also discuss this point in the response to reviewer PuV8.
>
> To our knowledge, the systematic, quantitative evidence on utility of synthetic data generated by T2I models over prompt complexity perspective is novel and has not been reported. We are happy to incorporate related works that have reported similar results into our manuscript if you have specific suggestions.

---

> ### Author Response · Authors · 2025-11-22
> **Response to Reviewer N9uF - Part 2**
>
> + **No concrete method to handle complexity, and question on using LLM-derived semantics**
>
> We appreciate the reviewer’s push toward a principled solution. Our work is primarily evaluative. However, we argue that **a rigorous diagnostic of the problem is a prerequisite for a principled solution**. By systematically quantifying how standard models behave with different prompt complexities, our work provides the necessary empirical foundation to design the training solutions the reviewer suggests.
>
> Regarding the specific question: “Could LLM-derived semantics be used during training?”
>
> Based on our findings, the answer is yes. Our results allow us to draw a direct connection between inference-time observations and training-time strategies:
>
> 1. **Reducing conditional ambiguity (Training Insight):** Incorporating LLM-derived semantics during training is a logical next step supported by our analysis. Short, general prompts create a high-entropy conditional distribution (mapping one text to a vast space of possible images), which is difficult for the model to learn. *Fine-grained, LLM-augmented detailed descriptions reduce this ambiguity, sharpening the conditional distribution and simplifying the learning objective.*
>
> 2. **Bridging the gap via inference:** Ideally, inference prompts should match the distribution of training prompts. However, because training data for large-scale models is often *unavailable or proprietary*, and retraining is computationally expensive, “training-time” solutions are not always accessible to practitioners. Our work demonstrates that *combining inference-time interventions* can approximate this training-inference alignment, improving the generation diversity and quality from T2I models.
>
> Thank you again for your detailed review and we are happy to discuss more if you have further questions.

---

### Official Review · Reviewer_QZqM · 2025-11-06

**Soundness:** 3
**Presentation:** 3
**Contribution:** 3
**Rating:** 6
**Confidence:** 3

**Summary:**

This paper investigates how prompt complexity affects synthetic data utility from T2I models across quality, diversity, and consistency axes. The authors (1) conduct synthetic experiments on Gaussian mixtures with theoretical derivations, (2) introduce an evaluation framework comparing real vs. synthetic data, and (3) perform large-scale empirical analysis across CC12M, ImageNet-1k, and DCI datasets with multiple T2I models and inference-time interventions. Key findings: increasing prompt complexity reduces diversity and consistency but narrows the real-synthetic gap; prompt expansion consistently improves diversity and quality; combining advanced guidance with prompt expansion yields optimal trade-offs

**Strengths:**

- **Important and underexplored research question.** This is the first systematic study of how prompt complexity affects T2I synthetic data utility. Given the widespread use of T2I models for data generation and the common practice of training on synthetic captions, understanding this relationship is timely and valuable.

- **Well-designed evaluation framework.** The 5-step pipeline (captioning, pairing, alignment, sampling, generation) enables fair comparisons between real and synthetic data across prompt complexities, a non-trivial methodological contribution that could benefit future work.

- **Comprehensive empirical evaluation.** The study covers multiple datasets (CC12M, ImageNet-1k, DCI), models (LDMv1.5/XL/v3.5M/v3.5L, Flux, Infinity), inference methods (CFG, CADS, Interval, APG, prompt expansion), and metrics (reference-free: Vendi, aesthetic, DSG; reference-based: FDD, precision, density, coverage), providing thorough coverage.

- **Human validation strengthens metric choice.** The human evaluation (Appendix E) shows Vendi score strongly correlates with human-perceived diversity, validating the automatic diversity metric.

- **Illustrative toy example provides intuition.** The synthetic Gaussian mixture experiments (Section 2) with mathematical derivations (Appendix A.2) offer clear intuition for why generalizing to general prompts is harder, nicely complementing the empirical findings.

**Weaknesses:**

- **Independence assumptions in theoretical derivations may not hold for real text encoders.** The synthetic experiments (Section 2, Appendix A.2) derive score functions assuming conditional independence of text concepts. Specifically, Equations 5-7 assume that for fine-grained conditioning $c_f$ composed of general concepts ${c^i_g}$, we have $p(x_t, c^1_g, c^2_g, ..., c^K_g) = p(x_t) ∏_{i} p(c^i_g|x_t)$. However, real text encoders (CLIP, T5) do not treat concepts as independent: (1) they encode compositional phrases where "white" modifying "dog" produces entangled representations rather than separable "white" and "dog" signals, (2) they learn correlations from training data where certain concept combinations (e.g., "white dog") appear frequently, making $p(c^1_g|x_t, c^2_g) ≠ p(c^1_g|x_t)$, and (3) concept embeddings are context-dependent and non-orthogonal in the latent space. Similarly, the OR operator derivation (Equation 1) requires fine-grained categories to be exhaustive and mutually exclusive, which may not align with how models internally represent general concepts. The Gaussian mixture model provides valuable intuition, but the theoretical predictions should be interpreted cautiously when applied to real T2I models. The authors should discuss this gap and consider how it might affect the interpretation of their results.
- **The alignment step lacks transparency and may introduce selection bias.** Algorithm 1 iteratively removes images not shared across complexities, but provides no analysis of what is discarded. Table 1 shows complexity-4 prompts cover only 46,066 images vs. 61,334 for complexity-1, a 25% reduction. What visual or semantic characteristics differentiate retained vs. discarded images? If alignment preferentially keeps images easier to describe with detailed prompts (e.g., clear objects vs. abstract scenes), this biases the evaluation set. Do discarded images have different mean aesthetic quality, diversity, or complexity than retained ones? Without this analysis, it is difficult to disentangle the genuine effects of prompt complexity from potential artifacts introduced by the evaluation pipeline itself.

**Questions:**

- **Can you characterize discarded images in the alignment step?** Specifically, do they have different mean aesthetic quality, object count, scene complexity, or semantic diversity compared to retained images? This would help assess whether the observed trends are artifacts of selection bias.
- **On Theoretical Assumptions:** Could you discuss the validity of the conditional independence assumption (Appendix A.2) in the context of compositional text encoders like CLIP? How might the violation of this assumption, where concepts like "red" and "car" are highly correlated and compositionally represented, affect the interpretation that generalizing to general prompts is inherently "harder" than generalizing to fine-grained ones? Could this theory-practice gap also explain some of the divergent behaviors observed between the CC12M and ImageNet experiments?
- **Can you provide confidence intervals or significance tests for main trends?** With 5,000 prompts per complexity, bootstrapped confidence intervals for Vendi, aesthetic, and FDD would strengthen claims about trends and help assess how statistically significant differences between adjacent complexity levels are.

---

> ### Author Response · Authors · 2025-11-22
> **Response to Reviewer QZqM - Part 1**
>
> Thank you very much for your insightful review, and for acknowledging our strengths, including the importance of the research question, the well-designed evaluation framework, the comprehensiveness of empirical results, the human evaluation and the illustrative toy synthetic example.
>
> + **Concern about the theoretical assumptions, including Weakness 1 and Question 2.**
>
> First, we would like to emphasize that we use the synthetic experiment as a **motivation** to emphasize the importance of studying the axis of **prompt complexity** in the diffusion / flow model setting. The synthetic experiments clearly show **an asymmetry of difficulty** w.r.t. conditioning complexity when the probabilistic assumptions hold. We do not intend to directly apply these theoretical predictions to real T2I models, but rather use them as an intuitive “clean room” that provides a hint when similar observations are witnessed in real settings.
>
> We acknowledge that the conditional independence assumption in our derivation is a simplification. Real-world text encoders (like CLIP/T5) produce **entangled representations** where concepts (e.g., “yellow” and “banana”) can be highly correlated and content dependant. However, we believe this distinction **does not invalidate** our conclusion regarding the asymmetry of difficulty between generalizing to general ("OR") vs. specific ("AND") prompts.
>
> 1. **On how this affects the “OR is harder than AND” conclusion:**
>
> **AND** Operator (Generalizing to longer prompts):
>
> In our toy example (Ideal Case): The “black” and “dog” concepts are independent. Their guidance vectors are orthogonal. Our Equation (2) is a probabilistically sound formula, and the naive sum works perfectly, as shown by the KL/FD scores (KL divergence of 0.93 and Frechet distance of 1.32).
>
> The violation of the assumption **doesn't make this operator "harder"** in the sense of being impractical, **but degrades it**. It changes Equation (2) from a probabilistic law into a practical but **imperfect heuristic**, probably leading to some failure modes. For example, the guidance vectors for “white” and “dog” are non-orthogonal if they are correlated as captured by the CLIP / T5 model. Our Equation (2), by naively adding these vectors, leads to an over-magnification of this shared, correlated signal, potentially leading to some artifacts (low diversity, over-saturation, etc.) in the generation.
>
> **OR** Operator (Generalizing to shorter prompts):
>
> This is already hard in theory as we explain in Eq. 1 in our manuscript, where the score function for "OR" operator is intractable in diffusion models. The fact that tokens in real-world prompts are not perfectly “mutually exclusive” adds another layer of intractability, making this “OR” generalization even more difficult.
>
> Given the theoretical analysis and not over-interpret our empirical results, some of our empirical results also verifies this asymmetry. For aesthetic quality with CC12M and DCI settings (Fig. 3, (a) and (c)), we observe a sharper decrease towards shorter prompts compared to longer prompts, which empirically supports our statement.
>
> 2. **On explaining CC12M vs. ImageNet behaviors:**
>
> Our theoretical analysis also helps explain some empirical divergences in results of CC12M and ImageNet. CC12M relies on **composition of tokens** (similar to the “AND” and “OR” operator logic), where we see similar trends as in the synthetic experiments. ImageNet experiments rely on **specificity** (using semantically richer tokens rather than more tokens). Because the ImageNet hierarchy does not strictly follow the combinatorial logic of these mathematical derivations, we do not observe the same sharper slope towards general prompts (Figs. 3 (a) and (b)).

---

> ### Author Response · Authors · 2025-11-22
> **Response to Reviewer QZqM - Part 2**
>
> **Concern about the selection bias, especially the alignment step, including Weakness 2 and Question 1**
>
> To address the concern regarding potential selection bias and artifacts, we performed a quantitative analysis **comparing the images Retained versus those Discarded** during the alignment process.
>
> We randomly sampled 10,000 images from both sets and evaluated them on **Aesthetic Quality** (using our paper’s model), **Object Count** (using YOLOv11n Ultralytics at $640\times640$ resolution), and **Semantic Diversity** (marginal Vendi Score using DINOv2 features). The results are presented in Table 1.
>
> The aesthetic quality is similar between retained and discarded images. The semantic diversity difference is small considering the Vendi Score range is [0, 10,000]. Object count shows a small gap with around 1 object difference. This evidence shows that the alignment step **does not drastically change the data distribution** and introduce much selection bias.
>
> Table 1: Potential selection bias from alignment
> |           | Aesthetic quality | Object count | Semantic diversity |
> | --------- | ----------------- | ------------ | ------------------ |
> | Retained  | 4.36±0.01         | 4.78±0.05    | 377.97±1.93        |
> | Discarded | 4.31±0.01         | 3.76±0.04    | 358.78±1.68        |
>
> We would like to explain a bit more on how we design our evaluation framework to **ensure a valid comparison across complexities**. The filtering consists of both pairing and alignment steps, and we strictly enforce an intersection constraint, that images being filtered out are the same across all complexity levels. As such, this is equivalent to the setting that given an image set, we describe each image with captions of different complexities. Thus, **the semantics across complexities are aligned**. However, there are still too many captions left to be used for synthetic data generation and for practicality. We therefore conduct the sampling step, making synthetic data generation in the evaluation feasible. Given the **large sample size** and the **shared semantics** across complexities, the semantic coverage across all complexities **remains statistically equivalent**. We believe that this ensures the validity of the trend observed.
>
> It is also worthwhile to distinguish between **Marginal Diversity** and **Conditional Diversity** (diversity of images generated from a specific caption, evaluated in our paper). While filtering may alter the marginal distribution of the original data, the primary focus of our study is conditional diversity – a property depending on the information level of a given prompt and the generative models. Since we **align the underlying semantics across complexities**, the observed trends in conditional diversity are generally **not artifacts in the data selection**.

---

> ### Author Response · Authors · 2025-11-22
> **Response to Reviewer QZqM - Part 3**
>
> **Question 3 Confidence Interval**
>
> We report the mean with 95% CI on CC12M in the following table. The mean and 95% CI are computed over the 5000 prompts. The confidence intervals (95% CI) are **narrow**, indicating **high stability** in the measurement. Notably, the intervals between consecutive complexity levels are **non-overlapping**, confirming that the main trends at each complexity are **statistically significant**.
>
> For diversity:
> | complexity             | 1            | 2            | 3            | 4           |
> | ---------------------- | ------------ | ------------ | ------------ | ----------- |
> | LDMv1.5 + CFG          | 9.228±0.094  | 7.649±0.079  | 6.957±0.067  | 6.483±0.059 |
> | +APG                   | 10.163±0.093 | 8.701±0.081  | 7.921±0.070  | 7.496±0.063 |
> | +CADS                  | 11.112±0.095 | 9.557±0.081  | 8.496±0.072  | 7.902±0.064 |
> | +Interval              | 11.934±0.089 | 10.378±0.082 | 9.314±0.070  | 8.839±0.066 |
> | +Prompt expansion, CFG | 15.074±0.089 | 12.977±0.097 | 10.882±0.092 | 9.725±0.081 |
> | LDMv3.5L               | 5.597±0.053  | 5.493±0.044  | 4.294±0.032  | 3.966±0.028 |
> | +APG                   | 8.978±0.090  | 7.331±0.079  | 6.089±0.065  | 5.302±0.054 |
> | +CADS                  | 9.864±0.108  | 7.690±0.093  | 5.954±0.067  | 5.214±0.056 |
> | +Interval              | 10.707±0.088 | 9.463±0.086  | 7.668±0.073  | 6.788±0.064 |
> | +Prompt expansion, CFG | 14.671±0.098 | 12.114±0.107 | 9.140±0.100  | 7.672±0.083 |
>
> For Quality:
> | complexity             | 1           | 2           | 3           | 4           |
> | ---------------------- | ----------- | ----------- | ----------- | ----------- |
> | LDMv1.5 + CFG          | 4.085±0.011 | 4.297±0.013 | 4.335±0.012 | 4.357±0.012 |
> | +APG                   | 4.035±0.011 | 4.213±0.012 | 4.214±0.011 | 4.233±0.011 |
> | +CADS                  | 4.013±0.011 | 4.195±0.012 | 4.239±0.011 | 4.254±0.011 |
> | +Interval              | 4.017±0.009 | 4.163±0.012 | 4.204±0.011 | 4.213±0.011 |
> | +Prompt expansion, CFG | 4.645±0.009 | 4.620±0.009 | 4.472±0.010 | 4.415±0.010 |
> | LDMv3.5L               | 5.011±0.011 | 5.155±0.012 | 5.016±0.013 | 4.981±0.013 |
> | +APG                   | 5.071±0.009 | 5.179±0.010 | 5.012±0.012 | 4.967±0.012 |
> | +CADS                  | 5.019±0.010 | 5.118±0.011 | 4.977±0.012 | 4.940±0.013 |
> | +Interval              | 5.042±0.008 | 5.101±0.009 | 4.987±0.009 | 4.924±0.011 |
> | +Prompt expansion, CFG | 5.238±0.008 | 5.229±0.009 | 5.110±0.010 | 5.072±0.010 |
>
> For consistency:
> | complexity             | 1             | 2             | 3             | 4             |
> | ---------------------- | ------------- | ------------- | ------------- | ------------- |
> | LDMv1.5 + CFG          | 0.9026±0.0026 | 0.8153±0.0029 | 0.7630±0.0025 | 0.7175±0.0022 |
> | +APG                   | 0.8853±0.0028 | 0.8011±0.0030 | 0.7520±0.0026 | 0.7036±0.0023 |
> | +CADS                  | 0.8724±0.0029 | 0.7928±0.0030 | 0.7438±0.0026 | 0.6981±0.0023 |
> | +Interval              | 0.8573±0.0030 | 0.7788±0.0031 | 0.7313±0.0026 | 0.6822±0.0023 |
> | +Prompt expansion, CFG | 0.8072±0.0034 | 0.7130±0.0034 | 0.6902±0.0028 | 0.6502±0.0024 |
> | LDMv3.5L               | 0.9235±0.0023 | 0.8404±0.0028 | 0.8001±0.0024 | 0.7622±0.0025 |
> | +APG                   | 0.9063±0.0025 | 0.8270±0.0029 | 0.7983±0.0024 | 0.7606±0.0021 |
> | +CADS                  | 0.8546±0.0031 | 0.7881±0.0031 | 0.7636±0.0025 | 0.7251±0.0022 |
> | +Interval              | 0.8677±0.0029 | 0.7901±0.0031 | 0.7715±0.0025 | 0.7378±0.0022 |
> | +Prompt expansion, CFG | 0.8473±0.0031 | 0.7494±0.0032 | 0.7432±0.0026 | 0.7083±0.0023 |
>
> Thank you again for acknowledging our strengths and for your insightful questions. We are happy to discuss if you have any further concern.

---

### Author Response · Authors · 2025-11-30
**General Response - Part I**

We thank the reviewers for their time and constructive feedback. We are encouraged by the strong consensus on the **significance and rigor** of our work. Reviewers highlighted that this is the **"first systematic study" of this "underexplored" and "interesting research question"** (QZqM, CGJq), commended the **"comprehensive, reproducible evaluation framework"** (N9uF, QZqM), and noted that the study provides **"crucial guidance" that sets an "important agenda for future T2I model development"** (PuV8). Reviewers also acknowledged the clarity and easy-to-follow nature of our presentation (N9uF, PuV8).

In this general response, we summarize our major updates and clarify our novelty and application contributions, specifically addressing the concern regarding "expected findings" and “limited novelty” by leveraging the consensus among reviewers. We also summarized the experiments provided during the rebuttal to ensure the robustness and validity of our evaluation results.

+ Novelty and Application value

A primary concern raised (N9uF, CGJq) was that some findings may align with intuition, limited novelty and application value. With our rebuttal, reviewer CGJq commented that "the concerns have been addressed".

We respectfully argue that **scientific progress requires moving from anecdotal "practitioner intuition" to rigorous, quantified evidence**. As Reviewer QZqM noted, this relationship is **"underexplored"**, and our work provides the necessary systematic verification, **enabled by the contributed benchmarking framework in our work**.

*To our knowledge, the systematic, quantitative evidence on the utility of synthetic data generated by T2I models from a prompt complexity perspective is novel and has not been reported.* In our responses (posted **a week earlier** than the no-more-discussion announcement) to reviewers who are concerned with the intuitive findings, we kindly asked that we would be very happy to incorporate related works that have reported similar results into our manuscript if they have specific suggestions.

Furthermore, our analysis revealed several **non-trivial and unexpected findings** — some of which described as "insightful observations" by Reviewer CGJq — that contradict common assumptions. We summarize these below:

(1) The trend of utilities are non-linear, and especially the aesthetic score exhibits a slope steeper towards shorter prompt lengths and more gradual for longer ones (Fig. 3 c), showing **an asymmetry of prompt length generalization**.

(2) Contrary to intuitions that very long prompts should collapse diversity, we find that diversity plateaus as prompt length increases, as shown in DCI (Fig. 2 c). This suggests **an inherent “lower bound of diversity”** in T2I models.

(3) We find that **optimizing for reference-free metrics harms distributional fidelity**. As shown in Figs. 5 and 13, prompt expansion degrades precision and density while newer models (e.g., LDMv3.5L) degrade in frechet distance with real data. This implies that optimizing for reference-free metrics may drive the generative distribution away from the real data manifold, suggesting **a cautious usage of synthetic data generated from T2I models for downstream applications**. We further discuss this point in the response to Reviewer PuV8.

Our evaluation results also lead to **actionable insights** for T2I model training and inference (Reviewer N9uF, CGJq).

1. **Reducing conditional ambiguity (Training Insight):** Incorporating LLM-derived semantics during training is a logical next step supported by our analysis. Short, general prompts create a high-entropy conditional distribution (mapping one text to a vast space of possible images), which is difficult for the model to learn. *Fine-grained, LLM-augmented detailed descriptions reduce this ambiguity, sharpening the conditional distribution and simplifying the learning objective.*

2. **Bridging the gap via inference:** Ideally, inference prompts should match the distribution of training prompts. However, because training data for large-scale models is often *unavailable or proprietary*, and retraining is computationally expensive, “training-time” solutions are not always accessible to practitioners. Our work demonstrates that *combining inference-time interventions* can approximate this training-inference alignment, improving the generation diversity and quality from T2I models.

---

> ### Author Response · Authors · 2025-11-30
> **General Response - Part II**
>
> + Strengthened Evaluation Robustness
>
> Reviewers praised the "methodologically rigorous" nature of our study (PuV8). To further strengthen this validity and address specific concerns, we performed additional experiments and presented more statistical results during the rebuttal. In particular:
>
> 1. **Statistical Significance** (Reviewer QZqM): We computed 95% confidence intervals for our main trends on CC12M. The intervals are narrow and non-overlapping between complexity levels, confirming that **the reported utility trends are statistically significant**.
>
> 2. **Human Validation of DSG** (Reviewer PuV8 and CGJq): Complimentary to the human evaluation over the diversity metric (vendi score) presented already in our paper, we further conducted a human evaluation to validate the DSG consistency metric on long prompts. We employed human annotators to verify both prompt decomposition and VQA answers, which are two stages of the DSG score computation. The automatic prompt decomposition achieved **95.54% accuracy** validated by humans. Further, the DSG scores computed using human-generated answers showed **a strong Pearson correlation (0.89)** with that using our automatic VQA models generated answers, confirming the reliability of our consistency findings.
>
> 3. **Quantifying Selection Bias** (Reviewer QZqM): We emphasize that our evaluation pipeline ensures **a commonly shared image semantics across different caption complexities**, reflecting the nature that images can be captioned into different levels of details. We also analyzed the images discarded during our alignment step following the reviewer’s suggestions. Comparing 10,000 retained vs. 10,000 discarded images, we found small differences in aesthetic quality (4.36 vs 4.31), object counts (only 1 object difference) and semantic diversity (5% change). This confirms that the trends showing in our evaluation results are not artifacts from the data selection pipeline.
>
> We updated the manuscript and marked the changes in red. We believe this rigorous benchmarking framework and the systematic quantitative evaluation results constitute a solid contribution to the community.

---

### Meta-Review · Area_Chair_YM59 · 2026-01-09

**Summary:**

The paper analyzes how the utility (quality, diversity, consistency) of images generated by a test-to-image model depend on the properties of the prompt. To do so, the paper introduces an evaluation procedure making heavy use of VLMs. The paper then shows application of the framework to several T2I models and several test-time guidance methods. The results are mostly intuitive, but there are also fairly surprising findings.

Based on the reviews, the authors’ rebuttal, and the paper itself, the main strengths and weaknesses are as follows.

Pros:
1. Important practically relevant topic
2. Well-designed evaluation framework. Strengthened by human evaluation.
3. Fairly thorough experiment with several models, datasets, inference/guidance methods, and metrics

Cons:
1. The Gaussian “toy” example is not very convincing: trading models from scratch just on a couple of prompts is very different from training a proper T2I model that has trained on a large amount of data, including compositional. So it’s not entirely clear to which degree this example is representative of real models.
2. Most conclusions are pretty expected
3. In many plots (e.g. Figures 3, 4) the y axis range is quite narrow, so not.a huge amount of variance depending on the prompt. This somewhat reduces the significance of those findings.
4. The paper focuses on analysis and not so much on solutions of discovered problems
5. Presentation could be better - plots are all of the same flavor and do not very clearly highlight the main findings.

Overall, the paper is fairly borderline, with its ups and downs. Given that this is the first systematic evaluation of the impact of prompt on T2I generation and that the experiments are fairly thorough, I recommend acceptance.

**Reviewer Concerns:**

- Novelty and value for practical applications -> addressed reasonably in the rebuttal in that there were no prior systematic studies and that there’s value in addressing such questions with a scientific study
- No unexpected findings -> partially addressed in the rebuttal in that the authors highlight a few findings that are not fully expected
- Applicability of the theory to real T2I models -> the authors addressed it to some degree, but I think the question still stands
- Confidence intervals -> provided
- Issues are highlighted but not solved -> it’s unfair to ask of this problem to solve everything

**Reviewer Scores:**

I wouldn’t expect a lot of change, maybe 1-2 reviewers by 1 point.

---

### Decision · Program_Chairs · 2026-01-26

Accept (Poster)